# Learning High-dimensional Gaussian Mixture Models via a Fourier Approach

## Abstract

In this paper, we address the challenge of learning high-dimensional Gaussian mixture models (GMMs), with a specific focus on estimating both the model order and the mixing distribution from i.i.d. samples. We propose a novel algorithm that achieves linear complexity relative to the sample size $n$, significantly improving computational efficiency. Unlike traditional methods, such as the method of moments or maximum likelihood estimation, our algorithm leverages Fourier measurements from the samples, facilitating simultaneous estimation of both the model order and the mixing distribution. The difficulty of the learning problem can be quantified by the separation distance $\Delta$ and minimal mixing weight $w_{\min}$. For stable estimation, a sample size of $\Omega\left(\frac{1}{w_{\min}^2 \Delta^{4K-4}}\right)$ is required for the model order, while $\Omega\left(\frac{1}{w_{\min}^2 \Delta^{4K-2}}\right)$ is necessary for the mixing distribution. This highlights the distinct sample complexities for the two tasks. For $D$-dimensional mixture models, we propose a PCA-based approach to reduce the dimension, reducing the algorithm's complexity to $O(nD^2)$, with potential further reductions through random projections. Numerical experiments demonstrate the efficiency and accuracy compared with the EM algorithm. In particular, we observe a clear phase transition in determining the model order, as our method outperforms traditional information criteria. Additionally, our framework is flexible and can be extended to learning mixtures of other distributions, such as Cauchy or exponential distributions.

## 1 Introduction

### 1.1 Background

The Gaussian Mixture Model (GMM) is a widely used statistical model that has found numerous applications in various fields, including machine learning, pattern recognition, data clustering, and image processing. It is a powerful tool for modeling complex data and signals originating from sub-populations or distinct sources. The GMM represents a probability distribution as a weighted sum of Gaussian components, each characterized by its mean and covariance matrix. Formally, each observation of the GMM follows:

$$\boldsymbol{x} \sim \sum_{i=1}^{K} w_i \mathcal{N}(\boldsymbol{\mu}_i, \boldsymbol{\Sigma}_i) \tag{1}$$

where $w_i$ is the mixing weight such that $w_i > 0$ and $\sum_{i=1}^{K} w_i = 1$. The mean and the covariance matrix of the $i$-th component are denoted as $\boldsymbol{\mu}_i$ and $\boldsymbol{\Sigma}_i$, respectively. For each sample $\boldsymbol{x}$, we can introduce a latent variable $z \in \{1, \cdots, K\}$, with the marginal distribution of $z$ specified by the mixing weights:

$$\mathbb{P}(z = i) = w_i.$$

Thus, the GMM can also be expressed conditionally as

$$\boldsymbol{x}|(z = i) \sim \mathcal{N}(\boldsymbol{\mu}_i, \boldsymbol{\Sigma}_i). \tag{2}$$

Given the i.i.d. samples drawn from the mixture distribution, the challenge is to learn the underlying model. Generally, there are three formulations for learning mixtures:

- *Clustering*: estimate the latent variable $z_j$ for each sample $\boldsymbol{x}_j$;
- *Parameter estimation*: estimate the weights $w_i$'s, means $\boldsymbol{\mu}_i$'s and covariance matrix $\boldsymbol{\Sigma}_i$'s up to a global permutation;
- *Density estimation*: estimate the probability density function of the GMM under specific loss functions.

Existing methodologies for clustering primarily rely on $k$-means, which seeks to minimize:

$$\underset{z_j, \boldsymbol{\mu}_i}{\arg\min} \sum_{j=1}^{n} \sum_{i=1}^{K} \mathbf{1}\left\{z_j = i\right\} \left\|\boldsymbol{x}_j - \boldsymbol{\mu}_i\right\|^2, \tag{3}$$

where $\mathbf{1}\left\{z_j = i\right\} = 1$ if $z_j = i$ otherwise 0. It is well-known that solving the $k$-means exactly in the general case is NP-hard, even for two clusters (see Aloise et al. (2009)). Various computationally tractable approximation approaches have been proposed, including the widely used Lloyd's algorithm (Lloyd (1982)), nonnegative matrix factorization (NMF) (see Paatero & Tapper (1994); He et al. (2011); Zhuang et al. (2023)), and semidefinite programming (SDP) (see Peng & Wei (2007)). Note that Lloyd's algorithm iterates a two-phase of re-assigning the samples to clusters and re-computing the cluster means until convergence. The perfect clustering of the mixture depends on the separation distance defined as:

$$\Delta := \min_{1 \le i < j \le K} \left\|\boldsymbol{\mu}_i - \boldsymbol{\mu}_j\right\|. \tag{4}$$

It has been shown in Ndaoud (2022) that the critical threshold for a perfect clustering of a two-component Gaussian mixture with a unified covariance matrix $\sigma^2 I$ in $p$-dimension is:

$$\Delta^2 = \sigma^2 \left(1 + \sqrt{1 + \frac{2p}{n \log n}}\right) \log n \tag{5}$$

Similar results are obtained for the $K$-component mixture model in Chen & Yang (2021).

Parameter estimation and density estimation benefit from a larger sample size, contrasting with the perfect clustering scenario (5). Existing methodologies for learning the mixture can be broadly categorized into the maximum-likelihood method and the moment-based method. The maximum likelihood method aims to maximize the likelihood of the given samples. The likelihood function is defined as

$$L(\boldsymbol{x}_j\text{'s}|w_i\text{'s}, \boldsymbol{\mu}_i\text{'s}, \boldsymbol{\Sigma}_i\text{'s}) = \prod_{j=1}^{n} \left(\sum_{i=1}^{K} w_i g(\boldsymbol{x}_j; \boldsymbol{\mu}_i, \boldsymbol{\Sigma}_i)\right),$$

where $g(\boldsymbol{x}; \boldsymbol{\mu}, \boldsymbol{\Sigma})$ is the probability density function of Gaussian distribution with mean $\boldsymbol{\mu}$ and covariance $\boldsymbol{\Sigma}$. Numerous iterative methods for optimization are proposed to seek the maximum or local maximum of the likelihood function. Among them, the most widely used one is the EM(Expectation-Maximization) Algorithm(Dempster et al. (1977)). The EM algorithm iterates a two-step operation to find a local maximum of the logarithm likelihood function, which may not necessarily be the ground-truth parameters. The Lloyd's algorithm can be regarded as a deterministic version of the EM algorithm. The moment-based methods date back to Pearson (1894). However, Pearson's method has practical limitations due to its sensitivity to moment selection and the instability of finding roots of high-degree polynomials. Various modifications of the method of moments are proposed, such as the Generalized Method of Moments(Hansen (1982)) and the Denoised Method of Moments(Wu & Yang (2020)). The Markov Chain Monte Carlo (MCMC) methods are also commonly used to generate parameter samples from the posterior distribution, with prominent samplers including the Metropolis method (Metropolis et al. (1953)). Additionally, relating to this paper, Fourier approach is proposed and utilized to learn the GMMs in Qiao et al. (2022); Liu & Zhang (2024).

It is worth noting that both the clustering via $k$-means and the parameter estimation by maximum likelihood and moment-based methods require the model order $K$ as an input. However, the model order is often unknown a priori, necessitating a method for determining the appropriate order for model learning. To address the challenge of model order selection, various statistical criteria and information-theoretic measures have been proposed. These include the Akaike Information Criterion (AIC, Akaike (1998)) and Bayesian Information Criterion (BIC, Schwarz (1978)). These

methods aim to balance model complexity and goodness of fit, providing a quantitative measure to evaluate the trade-off between model complexity and data fidelity. Bayesian approaches can also be used to determine the model order. The variational inference method proposed by Corduneanu & Bishop (2001) allows for model order determination by assigning appropriate prior distributions to the parameters and maximizing the variational posterior distribution.

## 1.2 Problem Setting and Main Contributions

Given $n$ independent samples drawn from a $D$-dimensional Gaussian mixture distribution with a unified covariance matrix:

$$\boldsymbol{x}_j \sim \sum_{i=1}^{K} w_i \mathcal{N}(\boldsymbol{\mu}_i, \boldsymbol{\Sigma}), \quad j = 1, \cdots n. \tag{6}$$

We assume that the covariance matrix $\boldsymbol{\Sigma} \in \mathbb{R}^{D \times D}$ is known as prior information. This scenario is referred to as the Gaussian location mixture if $\boldsymbol{\Sigma} = \sigma^2 \boldsymbol{I}$. We define the separation distance $\Delta$ and the minimal weight $w_{\min}$ of the model (6) as

$$\Delta = \min_{1 \leq i < i \leq K} \|\boldsymbol{\mu}_i - \boldsymbol{\mu}_j\|, \quad w_{\min} = \min_{1 \leq i \leq K} w_i.$$

In this paper, we focus on the parameter estimation of the mixture model (6). Specifically, we aim to determine the model order $K$ and estimate the mixing distribution $\nu(\boldsymbol{x}) = \sum_{i=1}^{K} w_i \delta_{\boldsymbol{\mu}_i}(\boldsymbol{x})$ of the model from the independent samples.

**Contributions:** We propose an efficient algorithm to estimate the model order and parameters simultaneously for high-dimensional GMMs, extending the previous work in Liu & Zhang (2024) for one dimension. The time complexity of the proposed algorithm is linear in the sample size $n$, making it highly scalable. The main novelty of our approach is the leverage of the Fourier measurements of the samples. This is naturally connected to the problem of super-resolution and of line spectral estimation, which can be solved efficiently using subspace methods such as the MUltiple SIgnal Classification (MUSIC) algorithm. To handle high-dimensional data, we apply Principal Component Analysis (PCA) to reduce the dimension to reduce the complexity to $O(D^2)$, significantly improving computational efficiency in large-scale applications. We compare our algorithm with the EM algorithm to highlight its advantages across different scenarios. We note that the Fourier approach in this paper differs from the one in Qiao et al. (2022), which primarily focuses on spherical GMMs in low-dimensional settings and is based on estimating the Fourier transform of the mixture at carefully chosen frequencies.

We establish a fundamental limit to estimating the model order and mixing distribution in the mixture model using the Fourier measurements. Specifically, we show that stable recovery of the model order requires a sample size of $n = \Omega\left(\frac{1}{w_{\min}^2 \Delta^{4K-4}}\right)$, while stable estimation of the means requires $n = \Omega\left(\frac{1}{w_{\min}^2 \Delta^{4K-2}}\right)$, respectively. This result quantifies the distinct sample complexities for these two tasks. We also provide multiple comparison tests with other model order estimation methods and illustrate a phase transition in the estimation accuracy.

## 1.3 Paper Organization and Notations

The rest of the paper is organized as follows. In Section 2, we propose Algorithm 1 for model order and mixing distribution estimation of GMMs and establish the sample size guarantee for stable estimation. In Section 3, we use PCA to reduce the time complexity of Algorithm 1 in high-dimensional mixtures. We performed several numerical experiments to illustrate the accuracy, resolution, and efficiency of the algorithms in Section 2.

Throughout the paper, we write $f(n) = O(g(n))$ if there exists some constant $c_1 > 0$ such that $f(n) < c_1 g(n)$, and $f(n) = \Omega(g(n))$ if there exists some constant $c_2 > 0$ such that $f(n) > c_2 g(n)$. We denote $f(n) \asymp g(n)$ if $f(n) = \Omega(g(n))$ and $f(n) = O(g(n))$. For a $k$-dimensional subspace $\boldsymbol{W}$ of $\mathbb{R}^n$, the projection of vector $\boldsymbol{v} \in \mathbb{R}^n$ on to $\boldsymbol{W}$ is defined as $\text{Proj}_{\boldsymbol{W}}(\boldsymbol{v}) = \arg\min_{\boldsymbol{u} \in \boldsymbol{W}} \|\boldsymbol{u} - \boldsymbol{v}\|_2$. $\boldsymbol{I}_D$ denotes the identity matrix of rank $D$.

## 2 OUR PROPOSAL: MODEL ORDER AND MIXING DISTRIBUTION ESTIMATION VIA FOURIER APPROACH

### 2.1 ALGORITHM

In this section, we present our algorithm for model order and mixing distribution estimation of high-dimensional GMMs. Our approach leverages the Fourier transform of the mixture distribution and highlights a natural connection with line spectral estimation (LSE) and super-resolution (SR). We assume that the means $\boldsymbol{\mu}_i \in [-R, R)^D$ for some $R > 0$. The probability density function of the distribution (6) can be expressed in a convolutional form:

$$p(\boldsymbol{x}) = g(\boldsymbol{x}; \boldsymbol{\Sigma}) * \sum_{i=1}^{K} w_i \delta_{\boldsymbol{\mu}_i}(\boldsymbol{x}) \tag{7}$$

where $g(\boldsymbol{x}; \boldsymbol{\Sigma})$ is the density function of the Gaussian distribution $\mathcal{N}(\boldsymbol{0}, \boldsymbol{\Sigma})$. We now consider the Fourier transform of (7):

$$\phi(\boldsymbol{t}) = \mathcal{F}[p(\boldsymbol{x})] = e^{-\boldsymbol{t}^{\mathrm{T}} \boldsymbol{\Sigma} \boldsymbol{t}} \sum_{i=1}^{K} w_i e^{\iota \langle \boldsymbol{\mu}_i, \boldsymbol{t} \rangle}, \tag{8}$$

where $\mathcal{F}[\cdot]$ denotes the Fourier transform. The function $\phi(\boldsymbol{t})$ is also known as the characteristic function (CF) of the mixture model in the context of the statistics. It can be estimated from the samples by the empirical characteristic function (ECF):

$$\psi_n(\boldsymbol{t}) = \frac{1}{n} \sum_{j=1}^{n} e^{\iota \langle \boldsymbol{x}_j, \boldsymbol{t} \rangle}. \tag{9}$$

According to the central limit theorem, the ECF follows asymptotic normality:

$$\sqrt{n} \left( \psi_n(\boldsymbol{t}) - \phi(\boldsymbol{t}) \right) \xrightarrow{d} \mathcal{N}(0, 1 - |\phi(\boldsymbol{t})|^2), \quad n \to +\infty.$$

By modulating (9) with the term $e^{\boldsymbol{t}^{\mathrm{T}} \boldsymbol{\Sigma} \boldsymbol{t}}$, we obtain:

$$e^{\boldsymbol{t}^{\mathrm{T}} \boldsymbol{\Sigma} \boldsymbol{t}} \psi_n(\boldsymbol{t}) = \sum_{i=1}^{K} w_i e^{\iota \langle \boldsymbol{\mu}_i, \boldsymbol{t} \rangle} + \epsilon_n(\boldsymbol{t}), \tag{10}$$

where the right-hand side consists of a linear combination of exponential signals and a noise term $\epsilon_n(\boldsymbol{t})$ that is due to the finite sample size $n$. The estimation of $\boldsymbol{\mu}_i$'s from the measurement (10) is known as the Line Spectral Estimation (LSE), see Stoica et al. (2005). Due to the exponential decay of the Fourier data $\phi(\boldsymbol{t})$, the available measurement in (10) is band-limited in the sense that there exist positive numbers $f_1, f_2, \cdots, f_D$, called cutoff frequencies, such that only measurement at $\boldsymbol{t} = (t_1, \cdots, t_D)$ with $|t_i| \leq f_i$ for $1 \leq i \leq D$ can be used for estimation. Estimating $\boldsymbol{\mu}_i$'s when they are closely separated from the band-limited Fourier data is a super-resolution problem, see Donoho (1992). The success of LSE depends crucially on the noise level and the cutoff frequencies. In our problem, the noise level $\|\epsilon_n(\boldsymbol{t})\|_\infty$ can be estimated quantitatively in a probabilistic manner by the following proposition:

**Proposition 1.** *For any fixed $\epsilon > 0$, we have*

$$\mathbb{P} \left( \left| e^{\boldsymbol{t}^{\mathrm{T}} \boldsymbol{\Sigma} \boldsymbol{t}} \psi_n(\boldsymbol{t}) - \sum_{i=1}^{K} w_i e^{\iota \langle \boldsymbol{\mu}_i, \boldsymbol{t} \rangle} \right| \geq \epsilon \right) \leq 4 \exp \left\{ -\frac{n \epsilon^2}{4 e^{2 \boldsymbol{t}^{\mathrm{T}} \boldsymbol{\Sigma} \boldsymbol{t}}} \right\} \leq 4 \exp \left\{ -\frac{n \epsilon^2}{4 e^{2 \|\boldsymbol{t}\|_2^2 \sigma_{\max}(\boldsymbol{\Sigma})}} \right\}$$

*where $\sigma_{\max}(\boldsymbol{\Sigma})$ denotes the maximal singular value of $\boldsymbol{\Sigma}$. Then for any $\delta \in (0, 1)$, if the sample size $n \geq 4 \log \left( \frac{4}{\delta} \right) \frac{e^{2 \|\boldsymbol{t}\|_2^2 \sigma_{\max}(\boldsymbol{\Sigma})}}{\epsilon^2}$, with probability $1 - \delta$, we have that*

$$\epsilon_n(\boldsymbol{t}) < \epsilon.$$

Given measurement (10) at a uniform grid of domain $[-f_1, f_1] \times \cdots [-f_D, f_D]$, we employ a MUSIC-type algorithm to estimate both the model order $K$ (i.e., the number of Gaussian components) and mixing distribution $\nu(\boldsymbol{x}) = \sum_{i=1}^{K} w_i \delta_{\boldsymbol{\mu}_i}(\boldsymbol{x})$. Introduced by Schmidt (1986), the MUltiple SIgnal Classification (MUSIC) algorithm is a widely utilized technique in frequency estimation,

spectral analysis, and radar signal processing, renowned for its high-resolution parameter estimation capabilities. Essentially, the MUSIC algorithm exploits the exponential form of the signals (similar to Prony's method introduced in Prony (1795)), as defined in Equation (10), to construct a Hankel matrix that admits a Vandermonde decomposition. The algorithm proceeds by performing Singular Value Decomposition (SVD) on the Hankel matrix to identify the noise subspace. Subsequently, it formulates an imaging function (denoted as $\mathcal{J}(\boldsymbol{\mu})$ in the algorithm) by computing a noise-space correlation function. In the noiseless scenario, the imaging function exhibits peaks precisely at the set of Gaussian means $\{\boldsymbol{\mu}_j\}_{1\le j\le K}$. In the presence of noise, the algorithm determines the number of Gaussian means by identifying the number of local maxima in the imaging function and estimates the set of means based on the locations of these maxima. The details of the MUSIC algorithm can be found in Appendix B. In Section 2.2 and 2.3, we discuss how to select the cutoff frequencies $f_1, \cdots, f_D$ and the number of sampling points to balance the computational tractability and estimation accuracy. The mixing weights are estimated using the quadratic programming, as detailed in Appendix F. We summarize the model order selection and mixing distribution estimation in Algorithm 1.

---

**Algorithm 1:** Model Order Selection and Mixing Distribution Estimation

**input** : samples $\boldsymbol{X}_1, \cdots, \boldsymbol{X}_n$, covariance matrix $\boldsymbol{\Sigma}$, cutoff frequencies $(f_1, \cdots, f_D)$, a prior upper bound for the number of Gaussian components $L$, sample size of the Fourier measurement in each direction $N$ with $N > L + K$.

1 Compute $y_n(\boldsymbol{t}) = e^{\boldsymbol{t}^\mathrm{T}\boldsymbol{\Sigma}\boldsymbol{t}}\psi_n(\boldsymbol{t})$ on the uniform grid of $[-f_1, f_1] \times \cdots \times [-f_D, f_D]$ with $(N+1)$ sample points along each direction;

2 Apply Algorithm 3 with input $y_n(\boldsymbol{t}), N, L$ and plot the imaging function $\mathcal{J}(\boldsymbol{\mu})$ in $[-R, R)^D$;

3 Return the model order $\hat{K}$ as the number of local maxima of $\mathcal{J}(\boldsymbol{\mu})$ and the means as the local maxima $\{\hat{\boldsymbol{\mu}}_i\}_{1\le i\le \hat{K}}$;

4 Return the weights $\{\hat{w}_i\}_{1\le i\le \hat{K}}$ by solving the quadratic programming problem (35);

**output:** estimated mixing distribution $\hat{\nu}(\boldsymbol{x}) = \sum_{i=1}^{\hat{K}} \hat{w}_i \delta_{\hat{\boldsymbol{\mu}}_i}(\boldsymbol{x})$.

---

We remark that this algorithm is also applicable when the model order $K$ is known. In that case, the means are determined by selecting the largest $K$ local maxima of the imaging function $\mathcal{J}(\boldsymbol{\mu})$.

## 2.2 PARAMETER SETUP OF ALGORITHM 1

In this section, we discuss how to set the parameters in Algorithm 1. Recall the Gaussian means $\boldsymbol{\mu}_j$'s are located within $[-R, R)^D$. By the the Nyquist–Shannon sampling theorem, the sampling step size $h$ for each direction in the Fourier domain should satisfy $0 < h \le \frac{\pi}{R}$, resulting in the following condition on the sample size $N$ in each direction:

$$N \ge \frac{2f_d R}{\pi}, \quad d = 1, \cdots, D.$$

We assume that we have a prior upper bound $L$ of the number of Gaussian components $K$ with $L = O(K)$. To recover $L$ components by the MUSIC algorithm, a sufficient condition on $N$ (see Appendix B) is:

$$N \ge 2L + 1.$$

Therefore

$$N = \max\left(2L + 1, \left\lceil \frac{2f_{\max} R}{\pi} \right\rceil\right), \tag{11}$$

where $f_{\max} = \max\{f_d : d = 1, \cdots, D\}$ and $\lceil x \rceil$ is the smallest integer greater or equal to $x$. The choice of cutoff frequencies will be discussed in Section 2.3.

## 2.3 TIME AND SAMPLING COMPLEXITY OF ALGORITHM 1

In this section, we analyze the time and sampling complexity of Algorithm 1 with parameters set as (11). We also propose a method for determining the cutoff frequencies. The time complexity of computing the Fourier measurement $y_n(\boldsymbol{t})$ is given by:

$$O\left(n(N+1)^D\right).$$

For multidimensional MUSIC, the primary computational cost arises from the singular value decomposition and the evaluation of the imaging function $\mathcal{J}(\boldsymbol{\mu})$. Suppose the number of grid points for evaluating $\mathcal{J}(\boldsymbol{\mu})$ is $M$ along each dimension. The time complexity of Algorithm 3 is:

$$O\left(\min\left\{(L+1)^{2D}(N-L+1)^D, (L+1)^D(N-L+1)^{2D}\right\} + (2M)^D\right).$$

Using the inputs from (11), the overall time complexity of Algorithm 1 becomes

$$O(n2^D K^D + K^{3D} + 2^D M^D) \tag{12}$$

which is linear in sample size $n$, but exponential in dimensionality $D$. This complexity can be reduced using the dimension reduction method introduced in Section 3.

Next, we examine the sampling complexity in relation to the separation distance $\Delta$ and the minimal mixing weight $w_{\min}$. The reliability of the estimation provided by Algorithm 1 depends on these two parameters, as well as the noise level $|\epsilon_n(\boldsymbol{t})|$ which is determined by the sample size $n$. This relationship is closely connected to the computational resolution limits established in Liu & Zhang (2021b) for one-dimensional and Liu & Zhang (2021a) for multi-dimensional LSE. Before presenting the main theorem, we introduce the concept of the computational resolution limit for multi-dimensional LSE. Consider the multi-dimensional Fourier measurement defined as

$$y(\boldsymbol{t}) = \sum_{i=1}^K w_i e^{\iota\langle\boldsymbol{\mu}_i,\boldsymbol{t}\rangle} + \epsilon(\boldsymbol{t}), \quad \boldsymbol{t} \in \mathbb{R}^D, \quad \|\boldsymbol{t}\|_2 \leq f. \tag{13}$$

Assume that $\|\epsilon(\boldsymbol{t})\|_\infty < \sigma$.

**Definition 1.** *Given the Fourier measurement $y(\boldsymbol{t})$ in (13), we say that the $\hat{\nu}(\boldsymbol{x}) = \sum_{i=1}^{\hat{K}} \hat{w}_i \delta_{\hat{\boldsymbol{\mu}}_i}(\boldsymbol{x})$ is a $\sigma$-admissible discrete measure of $y(\boldsymbol{t})$ if*

$$\|\mathcal{F}\hat{\nu}(\boldsymbol{t}) - y(\boldsymbol{t})\|_\infty < \sigma, \quad \forall \|\boldsymbol{t}\|_2 \leq f.$$

The set of $\sigma$-admissible measures characterizes all the possible solutions of the inverse problem from Fourier measurements $y(\boldsymbol{t})$. If there exists an admissible measure $\hat{\nu}$ with less than $K$ components, one may miss out one or more sources and therefore cannot estimate the model order correctly. This leads to the definition of the computational resolution limit for number detection.

**Definition 2.** *The computational resolution limit for number detection in $D$-dimensional space is defined as the smallest nonnegative number $\mathcal{R}_{D,K}$ such that for all $K$-component measure $\sum_{i=1}^K w_i \delta_{\boldsymbol{\mu}_i}, \boldsymbol{\mu}_i \in B^D_{\frac{(K-1)\pi}{2f}}(\boldsymbol{0})$ and the associated Fourier measurement $y(\boldsymbol{t})$ in (13), if*

$$\Delta = \min_{1\leq i < j \leq K} \|\boldsymbol{\mu}_i - \boldsymbol{\mu}_j\| \geq \mathcal{R}_{D,K}$$

*then there exists no $\sigma$-admissible measure consisting less than $K$ components with Fourier measurements $y(\boldsymbol{t})$.*

A quantitative characterization of $\mathcal{R}_{D,K}$ is provided in Appendix D. It can be shown that, up to two constants depending only on $D, K$, the limit takes the form

$$\mathcal{R}_{D,K} \asymp \frac{\pi}{f}\left(\frac{\sigma}{w_{\min}}\right)^{\frac{1}{2K-2}}. \tag{14}$$

This computational limit indicates that to accurately estimate the model order from the Fourier measurements (10) of the samples. The noise level $\|\epsilon_n(\boldsymbol{t})\|$ must be small enough such that $\mathcal{R}_{D,K} \leq \Delta$. The following theorem establishes a non-asymptotic lower bound for the sample size required to accurately recover the model order.

**Theorem 1.** *Consider the $D$-dimensional mixture model $\sum_{i=1}^K w_i \mathcal{N}(\boldsymbol{\mu}_i, \boldsymbol{\Sigma})$ with $\boldsymbol{\mu}_i \in B^D_{\frac{(K-1)\pi}{2f}}(\boldsymbol{0})$. For any $\delta \in (0,1)$, if the sample size $n$ satisfies that*

$$n \geq C_{K,D} \log\left(\frac{4}{\delta}\right) \frac{e^{2f^2 \sigma_{\max}(\boldsymbol{\Sigma})}}{w_{\min}^2 (f\Delta)^{4K-4}}. \tag{15}$$

*Then with probability $1 - \delta$, $\Delta \geq \mathcal{R}_{D,K}$ holds. Here $C_{K,D}$ is a constant only relying on $K$ and $D$.*

**Remark 1.** *For exact estimation of the model order $K$ using the Fourier measurements (10), the sample size should satisfy that*

$$n = \Omega\left(\frac{1}{w_{\min}^2 \Delta^{4K-4}}\right). \tag{16}$$

*This reveals the relation of the sample size $n$ with the mixture model itself explicitly.*

**Remark 2.** *The computational resolution limit for support recovery has also been established in Liu & Zhang (2021a). Following this theory, the sample complexity for estimating the means of a $K$-component GMMs with an error threshold less than $\Delta/2$, where $\Delta$ is the separation distance of the means, should satisfy*

$$n = \Omega\left(\frac{1}{w_{\min}^2 \Delta^{4K-2}}\right).$$

The computational resolution limit also sheds light on setting the cutoff frequencies in Algorithm 1. From Proposition 1, the noise level in (10) is amplified by a factor of $e^{\boldsymbol{t}^{\mathrm{T}} \boldsymbol{\Sigma} \boldsymbol{t}}$. For a one-dimensional mixture with variance $\sigma^2$, the noise level is amplified by $e^{f^2 \sigma^2}$. To minimize the computational resolution limit, a straightforward calculation leads to the optimal cutoff frequency set as $f^{\text{optimal}} = \sqrt{\frac{2K-2}{\sigma^2}}$. Therefore, we can set the cutoff frequencies as

$$f_d = \sqrt{\frac{2L-2}{\boldsymbol{e}_d^{\mathrm{T}} \boldsymbol{\Sigma} \boldsymbol{e}_d}}, \quad d = 1, \cdots, D, \tag{17}$$

if $K$ is unknown. Along with (11), these parameters are tested in detail in the numerical experiments shown in Section 4.

## 3 PCA-BASED DIMENSION REDUCTION

The time complexity of Algorithm 1 is exponential in the data dimension $D$ (see (12)). For a $K$-component mixture model, the means $\mathcal{S}$ lies on a subspace at most dimension $K$. If we can identify this subspace and project the samples onto it, the computational complexity of the model order estimation can be significantly reduced. In this section, we introduce a PCA-based method for dimension reduction. The idea is to first project the data onto a low-dimensional linear manifold using Principle Component Analysis (PCA) before running Algorithm 1. We demonstrate that this projection-based technique can also be used to estimate the mixing distribution. The PCA is based on the Singular Value Decomposition(SVD) of the data matrix:

$$\boldsymbol{X} = \begin{bmatrix} \boldsymbol{x}_1 & \cdots & \boldsymbol{x}_n \end{bmatrix}^{\mathrm{T}} \in \mathbb{R}^{n \times D}.$$

Assume that $n > D$ and denote its singular value decomposition as

$$\boldsymbol{X} = \sum_{d=1}^{D} \lambda_d \boldsymbol{u}_d \boldsymbol{v}_d^{\mathrm{T}}$$

where $\lambda_1 \geq \lambda_2 \geq \cdots \geq \lambda_D \geq 0$. Denote $\boldsymbol{V}_k = \begin{bmatrix} \boldsymbol{v}_1 & \cdots & \boldsymbol{v}_k \end{bmatrix} \in \mathbb{R}^{D \times k}$. The PCA projects the samples onto the column space of $\boldsymbol{V}_k$. We summarize the PCA-based model order and mixing distribution estimation algorithm below.

---

**Algorithm 2:** PCA-based Model Order Selection and Mixture Distribution Estimation

---

**input** : samples $\boldsymbol{x}_1, \cdots, \boldsymbol{x}_n, \boldsymbol{\Sigma}, (f_1, \cdots, f_k), k, N, L$
1 Compute the SVD of data matrix $X = \sum_{d=1}^{D} \lambda_d \boldsymbol{u}_d \boldsymbol{v}_d^{\mathrm{T}}$;
2 Project the samples to the subspace spanned by $\boldsymbol{v}_1, \cdots, \boldsymbol{v}_k$;
3 Run Algorithm 1 with inputs as projected samples, $\boldsymbol{V}^{\mathrm{T}} \boldsymbol{\Sigma} \boldsymbol{V}, (f_1, \cdots, f_k), N, L$ ;
4 Transfer the projected means into their original space;
**output:** the model order $\hat{K}$ and the mixing distribution.

---

If the model order $K$ is known, set $k = K$. If $k < K$, the projection may miss components lying orthogonal to the space spanned by the principle components. This issue and possible solutions are

discussed in Section C.2. In the next subsection, we provide a theoretical analysis of the dimension reduction using PCA. This method reduces the time complexity from (12) to

$$O\left(nD^2 + n2^k K^k + K^{3k} + 2^k M^k\right). \tag{18}$$

This reduces the exponential dependency on $D$ to $k$. If $K = O(1)$ relative to the sample size $n$ and dimensionality $D$, then the time complexity of Algorithm 2 becomes $O(nD^2)$, which is quadratic in the dimensionality. This complexity can be further reduced by random projection techniques. For instance, one can first apply the Johnson-Lindestrauss embedding to project the data onto a subspace of dimension $\Omega\left(\frac{\log K}{\epsilon^2}\right)$. The estimation accuracy remains promising if the shrunk separation distance $(1 - \epsilon)\Delta$ remains above the resolution limit.

### 3.1 ANALYSIS ON THE GAUSSIAN LOCATION MIXTURE

In this section, we consider the Gaussian mixture with covariance matrix as $\sigma^2 I$, also known as the Gaussian location mixture. We demonstrate that when $n > D$, the expected subspace spanned by the first $K$ right singular vectors $\{v_1, \cdots v_K\}$ in PCA either includes or coincides with the subspace spanned by the means $\{\mu_1, \cdots \mu_K\}$.

Firstly, recall that

$$\text{span}\{v_1, \cdots, v_K\} = \underset{\{V : \dim V = K\}}{\arg\max} \|\text{Proj}_V X\|_2. \tag{19}$$

We have the following theorem, where part of the proof is adapted from Vempala & Wang (2002):

**Theorem 2.** *Given any $k$-dimensional $(k \leq D)$ subspace spanned by orthonormal vectors* $\{w_1, \cdots w_k\}$*. Denote $W_k = \text{span}\{w_1, \cdots, w_k\}$ and $U_K = \text{span}\{\mu_1, \cdots, \mu_K\}$, then we have*

$$\mathbb{E}\left\|Proj_{U_K} X\right\|_2 \geq \mathbb{E}\left\|Proj_{W_k} X\right\|_2.$$

*Furthermore, if $k < K$, we have $\arg\max_{W_k} \mathbb{E}\left\|Proj_{W_k} X\right\|_2 \subset U_K$ and if $k \geq K$, we have* $U_K \subset \arg\max_{W_k} \mathbb{E}\left\|Proj_{W_k} X\right\|_2$.

This theorem implies that, with a sufficiently large sample size, the subspace obtained via SVD closely approximates the subspace spanned by the centers. Therefore, estimating the mixing distribution in the projected subspace is reasonable.

## 4 NUMERICAL RESULTS

### 4.1 COMPARISON WITH EM ALGORITHM

In this experiment, we compare the performance of Algorithm 2 with the EM algorithm for estimating the mixing distribution. We also include tests using PCA as a preprocessing step before applying the EM algorithm. All tests are conducted on a mixture model with dimension $D = 100$ and components $K = 2, 3$ with $\boldsymbol{\Sigma} = \boldsymbol{I}$.

The tests are designed as follows. For $K = 2$, samples are generated from the model $\frac{1}{2}\mathcal{N}(-\boldsymbol{\mu}, \boldsymbol{I}_{100}) + \frac{1}{2}\mathcal{N}(\boldsymbol{\mu}, \boldsymbol{I}_{100})$, and the mixture distribution is $\frac{1}{3}\mathcal{N}(-\boldsymbol{\mu}, \boldsymbol{I}_{100}) + \frac{1}{3}\mathcal{N}(\boldsymbol{0}, \boldsymbol{I}_{100}) + \frac{1}{3}\mathcal{N}(\boldsymbol{\mu}, \boldsymbol{I}_{100})$ for $K = 3$. In each test, the model order $K$ and the $\|\boldsymbol{\mu}\|_2$ are fixed. The sample size $n$ ranges from $10,000$ to $200,000$ with increments of $10,000$. With fixed $\|\boldsymbol{\mu}\|_2$, For each sample size, 96 independent trials are conducted, and the estimation error is averaged across trials. In each independent trial, the mean $\boldsymbol{\mu}$ is generated by first selecting a vector uniformly from the unit sphere $\mathbb{S}^{D-1}$, then scaling it by $\|\boldsymbol{\mu}\|_2$. The inputs for Algorithm 2 are set as $f = \sqrt{2K - 2}, k = K, L = K, N = 2K$ in accordance with (11) and (17). For the EM algorithm, the initial means are randomly set as $K$ samples, and the algorithm terminates after $5,000$ iterations or when the log-likelihood increases less than $1 \times 10^{-6}$. During the iterations of the EM algorithm, the covariance matrix is fixed as $\boldsymbol{I}_{100}$. The estimation error for the mixing distribution is defined using the Wasserstein distance:

$$W_1(\nu, \hat{\nu}) = \inf \mathbb{E}\left\|X - Y\right\|_2,$$

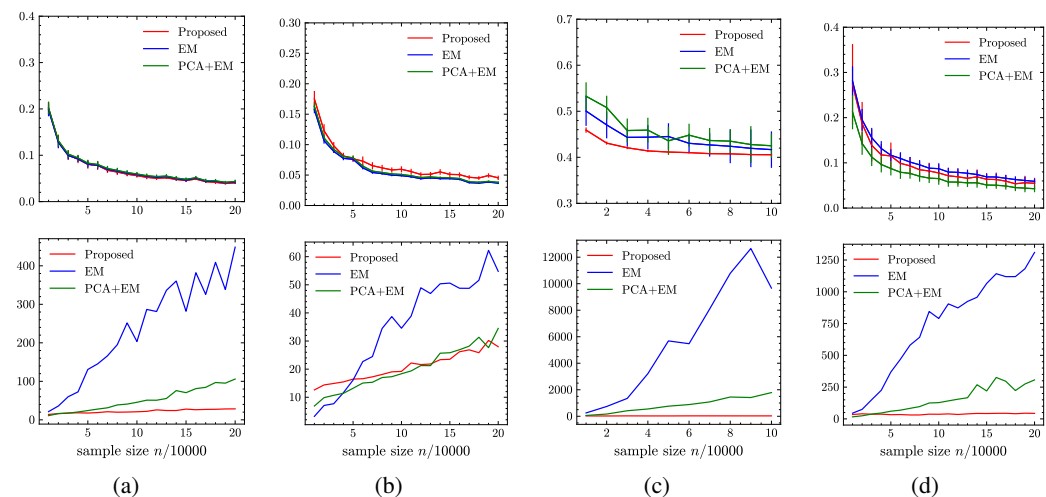

Figure 1: Comparison with the EM algorithm. The uppers are the $W_1$ errors of each method and the lowers are the average running time(seconds) of each trial. The samples of each trial comes from: (a) $\frac{1}{2}\mathcal{N}(-\boldsymbol{\mu}, \boldsymbol{I}_{100}) + \frac{1}{2}\mathcal{N}(\boldsymbol{\mu}, \boldsymbol{I}_{100}), \|\boldsymbol{\mu}\|_2 = 1$; (b)$\frac{1}{2}\mathcal{N}(-\boldsymbol{\mu}, \boldsymbol{I}_{100}) + \frac{1}{2}\mathcal{N}(\boldsymbol{\mu}, \boldsymbol{I}_{100}), \|\boldsymbol{\mu}\|_2 = 2$; (c)$\frac{1}{3}\mathcal{N}(-\boldsymbol{\mu}, \boldsymbol{I}_{100}) + \frac{1}{2}\mathcal{N}(\boldsymbol{0}, \boldsymbol{I}_{100}) + \frac{1}{3}\mathcal{N}(\boldsymbol{\mu}, \boldsymbol{I}_{100}), \|\boldsymbol{\mu}\|_2 = 1$; (d)$\frac{1}{3}\mathcal{N}(-\boldsymbol{\mu}, \boldsymbol{I}_{100}) + \frac{1}{2}\mathcal{N}(\boldsymbol{0}, \boldsymbol{I}_{100}) + \frac{1}{3}\mathcal{N}(\boldsymbol{\mu}, \boldsymbol{I}_{100}), \|\boldsymbol{\mu}\|_2 = 2$.

where the infimum is taken for all joint distributions of random vectors $(X, Y)$ with marginals $\nu, \hat{\nu}$ and this Wasserstein distance can be numerically computed through optimal transport [1]. The results are presented in Figure 1.

The results demonstrate that Algorithm 2 achieves comparable accuracy to the EM algorithm while requiring significantly less time, especially for the large sample sizes. This efficiency arises because the running time of Algorithm 2 scales linearly with the sample size $n$. In contrast, the EM algorithm is highly sensitive to the initialization of the means and the landscape of the likelihood function, which may result in slow convergence if the initialization or landscape is unfavorable. Additionally, the time complexity of the EM algorithm in each iteration is $O(nKD^2)$, which is approximately the total complexity order (18). The dependence of the convergence speed of the EM algorithm on the likelihood function's landscape is evident when comparing panels (a) with (b) and (c) with (d) in Figure 1. A larger separation distance results in a better landscape, leading to faster convergence for the EM algorithm.

## 4.2 RESOLUTION LIMIT AND PHASE TRANSITION OF MODEL ORDER ESTIMATION

In this experiment, we explore the resolution limit of Algorithm 1 and compare it with other commonly used model order estimation methods. Specifically, we test the resolution limit for equally weighted two-component and four-component mixture model in $\mathbb{R}^2$. The covariance matrices are $\boldsymbol{I}$ for all Gaussian components. The geometry of the component means is illustrated in Figure 2.

The tests are designed as follows. We uniformly take 2,800 $(\log_{10}(n), \Delta)$ points in the domain $[2.5, 6.0] \times [0.2, 6.0]$. For each $(\log_{10}(n), \Delta)$ pair, we construct the equally weighted mixture model with the means illustrated in Figure 2. We draw $n$ independent samples from the model and apply Algorithm 1, AIC, and BIC for model order estimation. For the $K$-component mixture, the inputs of Algorithm 1 are $f = \sqrt{K+1}, L = K+1, N = 2K+2$, which allows for the model order ranging from 1 to $K + 1$. For AIC and BIC, the model is estimated by the EM algorithm with model order ranging from 1 to $K + 1$. The EM algorithm terminates after $5,000$ iterations or the log likelihood increases less than $1 \times 10^{-5}$. The results are shown in Figure 2.

---

[1]In our experiments, we use `wasserstein_distance_nd` in the Python package `scipy`.

The results reveal phase transitions for all three methods. The proposed method demonstrates a more favorable phase transition region compared to the other two criteria. However, the transition may not be as pronounced as that of the information criteria. Further refinement of these criteria could enhance the performance of the proposed method.

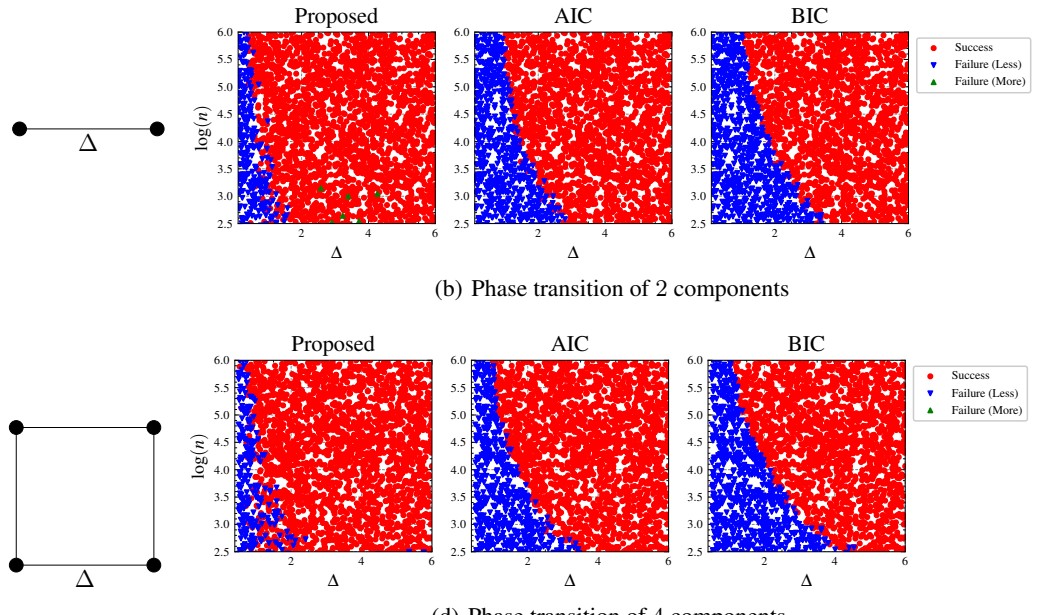

(b) Phase transition of 2 components

(d) Phase transition of 4 components

Figure 2: Geometry of the means and the phase transition of different model order estimation methods. The black dots stand for the mean locations and $\Delta$ stands for the separation distance. The blue triangle means the model order is underestimated and the green triangle means the model order is overestimated.

## 5 CONCLUSIONS AND DISCUSSIONS

Learning Gaussian mixture models is a challenging task, particularly in high dimensions or when the number of components is large or unknown. The performance of the learning algorithms depends on the separation distance and minimal weight of the components. In this paper, we proposed an efficient algorithm for estimating the model order and mixing distribution of the high-dimensional GMMs. Our algorithm leverages the Fourier measurement of the samples, drawing a natural connection to line spectral estimation and super-resolution techniques. We have established the sampling complexities for estimating the model order and mixing distributions in relation to the separation distance and minimal weight. Additionally, we demonstrated that the computational complexity for learning high-dimensional mixtures can be further reduced using dimension reduction techniques such as PCA. Empirical results confirmed that our algorithm achieves efficiency and accuracy comparable to, or better than, the EM algorithm.

We also acknowledge some aspects of our approach that present opportunities for future improvement. While our algorithm assumes that the unified covariance matrix $\Sigma$ is known a priori, there are scenarios where this may not be the case. To enhance the versatility of our method, estimating the covariance matrix using Fourier measurements, as explored in the 1-D algorithm in Liu & Zhang (2024), could be a promising direction. Additionally, while the time complexity remains quadratic with respect to dimensionality, this opens avenues for further research. Employing random projections that preserve pairwise distances between components, such as the Johnson–Lindenstrauss embedding (see Sanjeev & Kannan (2001)), could be an effective way to address this challenge. We will exploring these possibilities in future work.

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

## A    NOTATIONS

We shall use the following notations in the appendix. We use $\Re\left(\cdot\right)$ and $\Im\left(\cdot\right)$ to denote the real and imaginary part of a complex number, vector or matrix. For a matrix $\boldsymbol{A} \in \mathbb{R}^{m \times n}$(or $\mathbb{C}^{m \times n}$), we use $\boldsymbol{A}_j$ to denote the $j$-th column of $\boldsymbol{A}$ and the induced 2-norm $\|\boldsymbol{A}\|_2 = \sqrt{\sum_{j=1}^n \|\boldsymbol{A}_j\|_2^2}$.

## B    REVIEWS ON MUSIC ALGORITHM

In this section, we review the multidimensional MUltiple SIgnal Classification(MUSIC) algorithm. The MUSIC algorithm (see Schmidt (1986)) was initially proposed for the direction of arrival(DOA) detection and line spectral estimation(LSE). The multidimensional MUSIC is applied in $D$-dimensional single-snapshot spectral estimation. Consider a signal $y(t)$ which is a linear combination of $K$ time-harmonic components and additive noise $\epsilon(\boldsymbol{t})$:

$$y(\boldsymbol{t}) = \sum_{i=1}^K w_i e^{\iota \langle \boldsymbol{\mu}_i, \boldsymbol{t} \rangle} + \epsilon(\boldsymbol{t}). \tag{20}$$

The goal is to recover the frequency set $\mathcal{S} = \{\boldsymbol{\mu}_i : 1 \le i \le K\}$ and the corresponding amplitude $w_i$, from uniform samples of $y(\boldsymbol{t})$ in the domain $[-f, f]^D$. Suppose we have a total $(N+1)^D$ uniformly spaced sampling points with a grid size $h = \frac{2f}{N}$. Consequently, the frequencies can only be determined on the torus $\left[-\frac{N\pi}{2f}, \frac{N\pi}{2f}\right)^D$.

We first review the multidimensional MUSIC algorithm when $D = 2$. The extension to higher dimensions can be found in Liao (2015). For simplicity, we define the sampling coordinates along each direction as $t_q = -f + q\frac{2f}{N}$ for $q = 0, \cdots, N$, and $\boldsymbol{\mu}_i = (\mu_1^i, \mu_2^i)$ for $i = 1, \cdots, K$. We also introduce the following notation:

$$\phi_l(\mu) = \begin{bmatrix} 1 & e^{\iota \mu h} & \cdots e^{\iota \mu l h} \end{bmatrix}^{\mathrm{T}} \in \mathbb{C}^{l+1}.$$

Denote the noiseless uniform samples on the grid as:

$$y_{n_1,n_2} = \sum_{i=1}^K w_i e^{\iota \langle \boldsymbol{\mu}_i, \boldsymbol{t}_{n_1,n_2} \rangle}, \quad 0 \le n_1, n_2 \le N.$$

where $\boldsymbol{t}_{n_1,n_2} = (t_{n_1}, t_{n_2})$ is the sample point.

Given a fixed integer $L < N$, we form the Hankel matrix

$$\boldsymbol{A}_{n_1} = \begin{bmatrix} y_{n_1,0} & y_{n_1,1} & \cdots & y_{n_1,N-L} \\ y_{n_1,1} & y_{n_2,2} & \cdots & y_{n_1,N-L+1} \\ \vdots & \vdots & \ddots & \vdots \\ y_{n_1,L} & y_{n_1,L+1} & \cdots & y_{n_1,N} \end{bmatrix} \in \mathbb{C}^{(L+1)\times(N-L+1)}, \quad 0 \le n_1 \le N. \tag{21}$$

It is well known that $\boldsymbol{A}_{n_1}$ has the Vandermonde decomposition:

$$\boldsymbol{A}_{n_1} = \boldsymbol{\Phi}_{L,2} \boldsymbol{\Pi} \boldsymbol{\Lambda}_{n_1,1} \boldsymbol{\Phi}_{N-L,2}^{\mathrm{T}}, \tag{22}$$

where

$$\boldsymbol{\Phi}_{L,2} = \begin{bmatrix} \phi_L(\mu_2^1) & \phi_L(\mu_2^2) & \cdots & \phi_L(\mu_2^K) \end{bmatrix} \in \mathbb{C}^{(L+1)\times K},$$
$$\boldsymbol{\Pi} = \mathbf{diag}\left(w_1, w_2 \cdots, w_K\right),$$
$$\boldsymbol{\Lambda}_{n_1,1} = \mathbf{diag}\left(e^{\iota \mu_1^1 t_{n_1}}, e^{\iota \mu_1^2 t_{n_1}}, \cdots, e^{\iota \mu_1^K t_{n_1}}\right).$$

Next, we construct the 2-fold Hankel block matrix:

$$\boldsymbol{H} = \begin{bmatrix} \boldsymbol{A}_0 & \boldsymbol{A}_1 & \cdots & \boldsymbol{A}_{N-L} \\ \boldsymbol{A}_1 & \boldsymbol{A}_2 & \cdots & \boldsymbol{A}_{N-L+1} \\ \vdots & \vdots & \ddots & \vdots \\ \boldsymbol{A}_L & \boldsymbol{A}_{L+1} & \cdots & \boldsymbol{A}_N \end{bmatrix} \in \mathbb{C}^{(L+1)^2 \times (N-L+1)^2}. \tag{23}$$

For higher dimensions $D > 2$, the $D$-fold Hankel block matrix can be formed recursively as:

$$
\boldsymbol{H} = \begin{bmatrix}
\boldsymbol{A}_0 & \boldsymbol{A}_1 & \cdots & \boldsymbol{A}_{N_1-L_1} \\
\boldsymbol{A}_1 & \boldsymbol{A}_2 & \cdots & \boldsymbol{A}_{N_1-L_1+1} \\
\vdots & \vdots & \ddots & \vdots \\
\boldsymbol{A}_{L_1} & \boldsymbol{A}_{L_1+1} & \cdots & \boldsymbol{A}_{N_1}
\end{bmatrix},
$$

where $\boldsymbol{A}_l$ is the $(D-1)$-fold Hankel block matrix formed from the samples $\{y_{l,n_2,\cdots,n_D} : 0 \le n_2, \cdots, n_D \le N\}$.

For the 2-fold Hankel block matrix $\boldsymbol{H}$, it can be verified that

$$
\boldsymbol{H} = \begin{bmatrix}
\boldsymbol{\Phi}_{L,2}\boldsymbol{\Lambda}_{0,1} \\
\boldsymbol{\Phi}_{L,2}\boldsymbol{\Lambda}_{1,1} \\
\vdots \\
\boldsymbol{\Phi}_{L,2}\boldsymbol{\Lambda}_{L,1}
\end{bmatrix} \boldsymbol{\Pi} \begin{bmatrix} \boldsymbol{\Lambda}_{0,1}\boldsymbol{\Phi}_{N-L,2}^{\mathrm{T}} & \boldsymbol{\Lambda}_{1,1}\boldsymbol{\Phi}_{N-L,2}^{\mathrm{T}} & \cdots & \boldsymbol{\Lambda}_{N-L,1}\boldsymbol{\Phi}_{N-L,2}^{\mathrm{T}} \end{bmatrix}. \tag{24}
$$

Defining:

$$
\boldsymbol{\psi}_L(\boldsymbol{\mu}) = \mathrm{vect}\left(\left\{ e^{\iota\langle\boldsymbol{\mu},\boldsymbol{t}_{n_1,n_2}\rangle} : 0 \le n_1, n_2 \le L \right\}\right) \in \mathbb{C}^{(L+1)^2},
$$

the Vandermonde decomposition (24) can be written as:

$$
\boldsymbol{H} = \underbrace{\begin{bmatrix} \boldsymbol{\psi}_L(\boldsymbol{\mu}_1) & \cdots & \boldsymbol{\psi}_L(\boldsymbol{\mu}_K) \end{bmatrix}}_{\boldsymbol{\Psi}_L} \boldsymbol{\Pi} \begin{bmatrix} \boldsymbol{\psi}_{N-L}(\boldsymbol{\mu}_1) & \cdots & \boldsymbol{\psi}_{N-L}(\boldsymbol{\mu}_K) \end{bmatrix}^{\mathrm{T}}. \tag{25}
$$

In the noiseless case, we have the following result for recovering the frequencies (see Liao (2015)):

**Theorem 3.** *Suppose $\boldsymbol{\mu}_i \neq \boldsymbol{\mu}_j$ for all $1 \le i \neq j \le K$ and*

$$
L + 1 \ge K, \quad N - L + 1 \ge K. \tag{26}
$$

*Then we have $\mathrm{rank}\,(\boldsymbol{\Phi}_{L,2}) = \mathrm{rank}\,(\boldsymbol{\Phi}_{N-L,2}) = \mathrm{rank}\,(\boldsymbol{H}) = K$. Furthermore, for any $\boldsymbol{\mu} \in \left[-\frac{Nw}{2f}, \frac{Nw}{2f}\right)^2$, if (26) holds, we have*

$$
\boldsymbol{\mu} \in \mathcal{S} \Longleftrightarrow \boldsymbol{\psi}_L(\boldsymbol{\mu}) \in Im(\boldsymbol{\Psi}_L), \tag{27}
$$

*where $Im(\boldsymbol{\Psi}_L)$ is the column space of $\boldsymbol{\Psi}_L$.*

This theorem provides a criterion (27) for detecting the frequencies in the noiseless case. When the measurement is contaminated with noise $\epsilon(\boldsymbol{t})$, we can apply the MUSIC algorithm by performing Singular Value Decomposition(SVD) on $\boldsymbol{H}^\epsilon$:

$$
\boldsymbol{H}^\epsilon = \begin{bmatrix} \boldsymbol{U}_1^\epsilon & \boldsymbol{U}_2^\epsilon \end{bmatrix} \mathbf{diag}\,(\sigma_1^\epsilon, \cdots, \sigma_K^\epsilon, \cdots) \begin{bmatrix} \boldsymbol{V}_1^\epsilon & \boldsymbol{V}_2^\epsilon \end{bmatrix}^*, \tag{28}
$$

where $\boldsymbol{U}_1^\epsilon \in \mathbb{C}^{(L+1)^2 \times K}, \boldsymbol{U}_2 \in \mathbb{C}^{(L+1)^2 \times \min\{(L+1)^2,(N-L+1)^2\}-K}$ and $Im(\boldsymbol{U}_1^\epsilon), Im(\boldsymbol{U}_2^\epsilon)$ are called signal space and noise space, respectively. The algorithm is realized by projecting $\boldsymbol{\psi}_L(\boldsymbol{\mu})$ onto the noise space and drawing the MUSIC imaging function defined as:

$$
\mathcal{J}(\boldsymbol{\mu}) = \frac{\|\boldsymbol{\psi}_L(\boldsymbol{\mu})\|_2}{\|\boldsymbol{U}_2^{\epsilon*}\boldsymbol{\psi}_L(\boldsymbol{\mu})\|_2}. \tag{29}
$$

In the noiseless case, we have the relation that

$$
\boldsymbol{\mu} \in \mathcal{S} \Longleftrightarrow \mathcal{J}(\boldsymbol{\mu}) = \infty.
$$

In the noisy case, the frequency set $\mathcal{S}$ is determined by locating the local maxima of the imaging function $\mathcal{J}(\boldsymbol{\mu})$. The algorithm is summarized in Algorithm 3.

When the $K = |\mathcal{S}|$ is unknown, the MUSIC can also be applied by setting $K$ to some integer larger than $|\mathcal{S}|$ in Algorithm 3. In such cases, the frequency set is determined by identifying the local maxima of $\mathcal{J}(\boldsymbol{\mu})$ using appropriate criteria to avoid numerical instabilities. In our experiments, we simply use the criterion that the amplitude of the local maxima $\hat{\boldsymbol{\mu}}$ is larger than a preset threshold $w > 0$.

---

**Algorithm 3:** multidimensional MUSIC

---

**input** : $y^\epsilon(\boldsymbol{t})$ sampled on $[-f, f]^D$ with $(N+1)^D$ sample points, $K, L$

**1** Form the Hankel block matrix $\boldsymbol{H}^\epsilon \in \mathbb{C}^{(L+1)^D \times (N-L+1)^D}$ ;

**2** Perform the SVD: $\boldsymbol{H}^\epsilon = [\boldsymbol{U}_1^\epsilon \quad \boldsymbol{U}_2^\epsilon] \, \mathbf{diag}\,(\sigma_1^\epsilon, \cdots, \sigma_K^\epsilon, \cdots) \, [\boldsymbol{V}_1^\epsilon \quad \boldsymbol{V}_2^\epsilon]^*$ where
$\boldsymbol{U}_1 \in \mathbb{C}^{(L+1)^D \times K}$;

**3** Compute the MUSIC imaging function $\mathcal{J}(\boldsymbol{\mu})$ on the $\left[-\frac{N\pi}{2f}, \frac{N\pi}{2f}\right]^D$ ;

**output:** $\mathcal{S} = \{K \text{ largest local maxima of } \mathcal{J}(\boldsymbol{\mu})\}$

---

## C    Complements to Numerical Results

### C.1    Capacity of Learning Models with Large Model Order

In this experiment, we perform two numerical tests to illustrate the capacity of learning mixture model with a large model order in Algorithm 1 and 2. We first examine a 2-dimensional example of a 12-component mixture model with a unified covariance matrix $0.3\boldsymbol{I}$. Using $1,000$ samples from this distribution, we compare the performance of Algorithm 1 and the EM algorithm in estimating the component means. The EM algorithm is initialiezd with samples uniformly drawn from the data and terminates when the log-likelihood increases less than $1 \times 10^{-6}$. The inputs of Algorithm 1 are set as $f_1 = f_2 = 3, L = 12, N = 25$. The results are shown in Figure 3. In the figure, the true Gaussian components are illustrated as the red circles centered at the component mean with a radius $1.5$ times standard deviation, while the estimated ones are illustrated with the green dashed circle. We observe that with the specific initialization used, the EM converges in 292 iterations but gets trapped in a local maxima. Algorithm 1 does not suffer from initialization issues and provides a more accurate estimate of the mixture means.

Next, we consider a similar 12-component model but in a 100-dimensional space. The mixture means from Figure 3 are embedded into the $\mathbb{R}^{100}$ and each mean is perturbed by a Gaussian vector drawn from $\mathcal{N}(\boldsymbol{0}, 0.1\boldsymbol{I}_{100})$. We apply Algorithm 2 and the EM algorithm to estimate the mixture means in this high-dimensional setting. The results are shown in Figure 4. For visualization purposes, the estimates are projected onto the first two dimensions. It can also be seen from the estimation error that the Algorithm 2 outperforms the EM algorithm under this setting. Furthermore, when considering only the 1-Wasserstein error in the first two dimensions, the Algorithm 2 shows significantly better performance, with an error of $0.180$ compared to the EM algorithm's error of $0.439$.

### C.2    Projection: Issues and Potential Solutions

So far, we have focused on the numerical examples where the component means lie on or near a 2-dimensional subspace. However, a 2-dimensional projection may yield inaccurate estimations if the component means are distributed across a higher-dimensional space. The following experiment illustrates this issue. In this experiment, we consider a 6-component mixture model in $\mathbb{R}^3$. The mixture means are $\{\pm R\boldsymbol{e}_1, \pm R\boldsymbol{e}_2, \pm R\boldsymbol{e}_3\}$ where $R = 4$ and the covariance matrix is $\boldsymbol{I}_3$. We draw $2,000$ samples from this mixture model and use Algorithm 2 with $k = 2$ to estimate the mixing distribution. The estimation results seem reasonable when examining the estimated means projected onto the first two principal components. However, the accuracy degrades when considering the estimated means in the original $\mathbb{R}^3$ space. This discrepancy arises because, in this model, the first two principal components span a subspace close to $\mathrm{span}\{\boldsymbol{e}_1, \boldsymbol{e}_2\}$, making it challenging to accurately estimate components whose means lie along the $z$-axis. As a result, projecting only onto the subspace $\mathrm{span}\{\boldsymbol{v}_1, \boldsymbol{v}_2\}$ makes it impossible to accurately estimate the third component.

To address this issue, one potential solution is to project the samples onto a higher-dimensional subspace and estimate the projected means. makes it impossible to accurately estimate the third component. As shown in (12), the time complexity of the $D$-dimensional MUSIC algorithm is exponential with respect to the data dimension $D$. Alternative multidimensional line spectral methods (e.g. Sarkar & Pereira (1995); Tang et al. (2014); Fei & Zhang (2023)) could also be applied, but they may encounter similar challenges. Another approach is to project the samples onto alternative

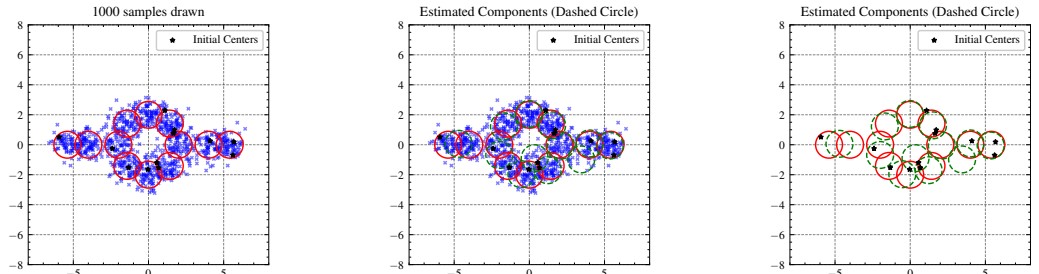

(a) Illustration of the EM algorithm with random initialization. Left: $1,000$ samples(blue cross) drawn from a 12-component mixture model and the initialization means(black star) of the EM algorithm (converges in 292 iterations); Middle: the estimated components by the EM algorithm; Right: the estimated components by the EM algorithm (without samples illustrated).

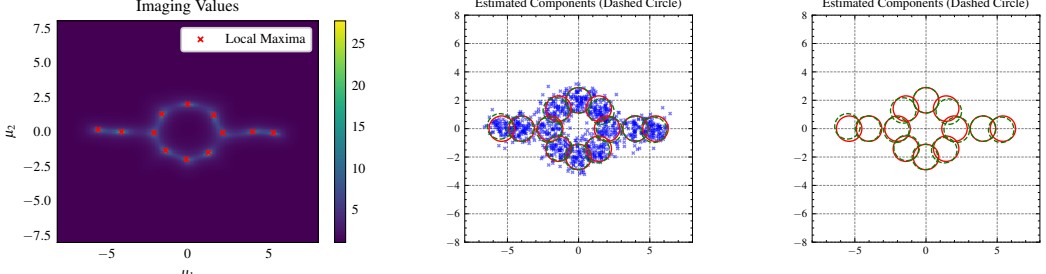

(b) Illustration of the Algorithm 1. Left: imaging function values of the MUSIC algorithm and the 12 largest local maximal; Middle: the estimated components by the Algorithm 1; Right: the estimated components by the Algorithm 1 (without samples illustrated).

Figure 3: Comparison of the EM algorithm and Algorithm 1 on the 12-component mixture model.

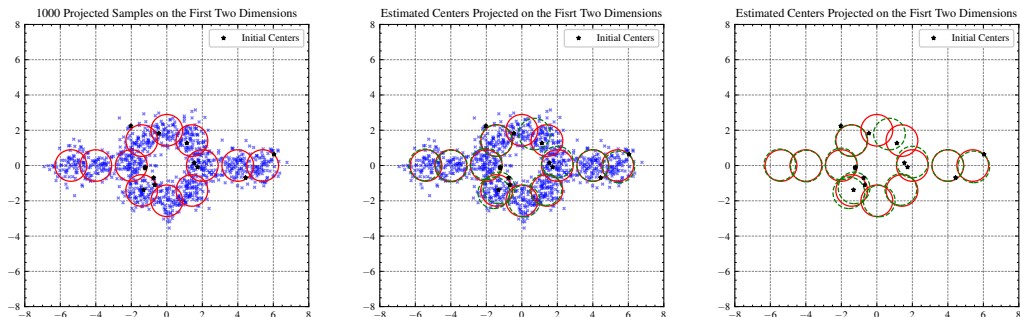

(a) Illustration of the EM algorithm with random initialization. Left: $1,000$ samples(blue cross) drawn from a 12-component mixture model and the initialization means(black star) of the EM algorithm (converges in 80 iterations); Middle: the estimated components by the EM algorithm; Right: the estimated components by the EM algorithm (without samples illustrated).

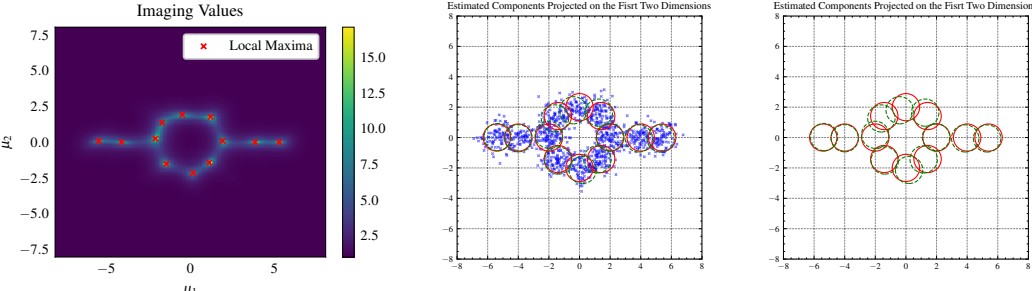

(b) Illustration of the Algorithm 2. Left: imaging function values of the MUSIC algorithm and the 12 largest local maximal; Middle: the estimated components by the Algorithm 1; Right: the estimated components by the Algorithm 1 (without samples illustrated).

Figure 4: Comparison of the EM algorithm and Algorithm 2 on the 12-component mixture model in $\mathbb{R}^{100}$.

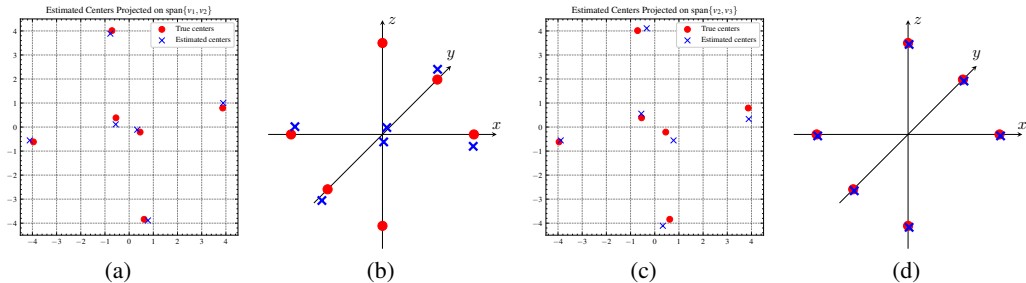

Figure 5: Geometry of the means. The black dots stand for the mean locations and $\Delta$ stands for the minimum distance.

2-dimensional subspaces that include the omitted principal directions. For example, in this case, we can project onto $\text{span}\{\boldsymbol{v}_2, \boldsymbol{v}_3\}$, and the results are shown in (c). A pairing process is necessary to reconstruct the estimated means under the basis $\{\boldsymbol{v}_1, \boldsymbol{v}_2, \boldsymbol{v}_2\}$. In this example, we simply pair the means whose $\boldsymbol{v}_2$ coordinates are closest and the estimation results are shown in (d). A more general pairing process might involve selecting the candidate with the highest likelihood. With this second projection procedure, we are able to accurately reconstruct along the $z$-axis, leading to improved estimation. This procedure can be generalized into the following algorithm.

---

**Algorithm 4:** Multiple Projections

---

**input** : samples $\boldsymbol{x}_1, \cdots, \boldsymbol{x}_n, \boldsymbol{\Sigma}$, number of projections $T, (f_1, \cdots, f_k), N, L$

**1** Compute the SVD of data matrix $\boldsymbol{X} = \sum_{d=1}^{D} \lambda_d \boldsymbol{u}_d \boldsymbol{v}_d^{\mathrm{T}}$;

**2 for** $i = 1, \cdots, T$ **do**

**3**     Project the samples on the the subspace spanned by $\boldsymbol{v}_i, \boldsymbol{v}_{i+1}$;

**4**     Run Algorithm 1 with inputs as projected samples,
        $\begin{bmatrix} \boldsymbol{v}_i & \boldsymbol{v}_{i+1} \end{bmatrix}^{\mathrm{T}} \boldsymbol{\Sigma} \begin{bmatrix} \boldsymbol{v}_i & \boldsymbol{v}_{i+1} \end{bmatrix}, (f_i, f_{i+1}), N, L$;

**5 end**

**6** Pair the projected estimations from each iteration above;

**7** Report the model order $\hat{K}$ and the mixing distribution.

---

## D    DETAILS ON SAMPLING COMPLEXITY

### D.1    COMPUTATIONAL RESOLUTION LIMIT

Consider the Fourier measurements of the high-dimensional line spectral signal as (13) and assume that $\|\epsilon(\boldsymbol{t})\|_\infty < \sigma$. The following theorem gives an upper bound for the computational resolution limit for the number detection:

**Theorem 4.** *(Liu & Zhang (2021a), Theorem 2.3) Let the Fourier measurement (13) be generated by an $n$-sparse measure $\nu = \sum_{i=1}^{K} w_i \delta_{\boldsymbol{\mu}_i}, \boldsymbol{\mu}_i \in B_{\frac{(K-1)w}{2f}}^{D}(\boldsymbol{0})$. Let $K \geq 2$ and assume the following separation condition is satisfied*

$$\Delta = \min_{1 \leq i < j \leq K} \|\boldsymbol{\mu}_i - \boldsymbol{\mu}_j\| \geq \frac{C_2(K, D)}{f} \left( \frac{\sigma}{w_{\min}} \right)^{\frac{1}{2K-2}} \tag{30}$$

*where $C_2(K, D) = 4.4we(w/2)^{s-1}(K(K-1)/w)^{\xi(s-1)}$ with*

$$\xi(k) = \begin{cases} \sum_{i=1}^{k} \frac{1}{i}, & k \geq 1 \\ 0, & k = 0, \end{cases}$$

*and $s$ being the the dimension of the smallest subspace in $\mathbb{R}^D$ which contains the set $\{\boldsymbol{\mu}_1, \cdots, \boldsymbol{\mu}_K\}$. Then there do not exist any $\sigma$-admissible measures of $y(\boldsymbol{t})$ with less than $n$ components.*

This theorem provides an upper bound of $\mathcal{R}_{K,D}$. The lower bound has also been characterized by the following proposition:

**Proposition 2.** *(Liu & Zhang (2021a), Proposition 2.4) For given $0 < \sigma < w_{\min}$ and $K \geq 2$, there exist an $K$-sparse measure in $\mathbb{R}^D$, $\nu = \sum_{i=1}^{K} w_i \delta_{\boldsymbol{\mu}_i}$ and an $(n-1)$-sparse measure in $\mathbb{R}^D$, $\hat{\nu} = \sum_{i=1}^{K-1} \hat{w}_i \delta_{\hat{\boldsymbol{\mu}}_i}$, such that $\|\mathcal{F}\hat{\nu}(\boldsymbol{t}) - \mathcal{F}\nu(\boldsymbol{t})\|_\infty < \sigma, \|\boldsymbol{t}\|_2 \leq f$. Moreover*

$$\min_{1 \leq i \leq K} |w_i| = w_{\min}, \quad \min_{1 \leq i < j \leq K} \|\boldsymbol{\mu}_i - \boldsymbol{\mu}_j\|_2 = \frac{C_1(K,D)}{f} \left( \frac{\sigma}{w_{\min}} \right)^{\frac{1}{2K-2}},$$

*where $C_1(K,D) = 0.81 e^{-\frac{3}{2}}$.*

The above results indicate that

$$\frac{C_1(K,D)}{f} \left( \frac{\sigma}{w_{\min}} \right)^{\frac{1}{2K-2}} < \mathcal{R}_{K,D} \leq \frac{C_2(K,D)}{f} \left( \frac{\sigma}{w_{\min}} \right)^{\frac{1}{2K-2}}. \tag{31}$$

The computational resolution limit for the support recovery of (13) has also been established in Liu & Zhang (2021a). Denote the computational resolution limit for support recovery as $\tilde{\mathcal{R}}_{K,D}$ and the results indicate that

$$\frac{\tilde{C}_1(K,D)}{f} \left( \frac{\sigma}{w_{\min}} \right)^{\frac{1}{2K-1}} < \tilde{\mathcal{R}}_{K,D} \leq \frac{\tilde{C}_2(K,D)}{f} \left( \frac{\sigma}{w_{\min}} \right)^{\frac{1}{2K-1}}, \tag{32}$$

where $\tilde{C}_1(K,D) = 0.49 e^{-\frac{3}{2}}$ and $\tilde{C}_2(K,D) = 5.88\pi e 4^{D-1} ((K+2)(K-1)/2)^{\xi(D-1)}$. From (31) and (32), it reveals the difference between these two tasks quantitatively by the $\frac{1}{2K-2}$ and $\frac{1}{2K-1}$ powered on the signal noise ratio term $\sigma/w_{\min}$.

### D.2 PROOF OF THEOREM 1

*Proof.* By setting $\epsilon = w_{\min} \left( \frac{\Delta f}{C_2(K,D)} \right)^{2K-2}$ in Proposition 1, we see that for

$$n \geq C_{K,D} \log\left(\frac{4}{\delta}\right) \frac{e^{2f^2 \sigma_{\max}(\boldsymbol{\Sigma})}}{w_{\min}^2 (f\Delta)^{4K-4}},$$

where $C_{K,D} = 4 \left(C_2(K,D)\right)^{4K-4}$, we have

$$\|\epsilon_n(\boldsymbol{t})\|_\infty \leq w_{\min} \left( \frac{\Delta f}{C_2(K,D)} \right)^{2K-2}, \|\boldsymbol{t}\|_2 \leq f$$

with probability at least $1 - \delta$. The rest follows from Theorem (4).

$\square$

Similar to Theorem 1, the sample size requirement for estimating the means is given by

**Theorem 5.** *Consider the $D$-dimensional mixture model $\sum_{i=1}^{K} w_i \mathcal{N}(\boldsymbol{\mu}_i, \boldsymbol{\Sigma})$ with $\boldsymbol{\mu}_i \in B_{\frac{(K-1)w}{2f}}^{D}(\boldsymbol{0})$. For any $\delta \in (0,1)$, if the sample size $n$ satisfies that*

$$n \geq \tilde{C}_{K,D} \log\left(\frac{4}{\delta}\right) \frac{e^{2f^2 \sigma_{\max}(\boldsymbol{\Sigma})}}{w_{\min}^2 (f\Delta)^{4K-2}}. \tag{33}$$

*Then with probability $1 - \delta$, $\Delta \geq \tilde{\mathcal{R}}_{D,K}$ holds. Here $C_{K,D}$ is a constant only relying on $K$ and $D$.*

The proof of the theorem is the same as that of Theorem 1.

## E  PROOF OF PROPOSITION 1

*Proof.* Note that

$$e^{\boldsymbol{t}^\mathrm{T}\boldsymbol{\Sigma}\boldsymbol{t}}\psi_n(\boldsymbol{t}) = \frac{1}{n}\sum_{j=1}^n e^{\boldsymbol{t}^\mathrm{T}\boldsymbol{\Sigma}\boldsymbol{t}}\cos\langle\boldsymbol{x}_j,\boldsymbol{t}\rangle + \iota\frac{1}{n}\sum_{j=1}^n e^{\boldsymbol{t}^\mathrm{T}\boldsymbol{\Sigma}\boldsymbol{t}}\sin\langle\boldsymbol{x}_j,\boldsymbol{t}\rangle.$$

Applying the Hoeffding's inequality to the real and imaginary parts, we have

$$\mathbb{P}\left(\left|\frac{1}{n}\sum_{j=1}^n e^{\boldsymbol{t}^\mathrm{T}\boldsymbol{\Sigma}\boldsymbol{t}}\cos\langle\boldsymbol{x}_j,\boldsymbol{t}\rangle - e^{\boldsymbol{t}^\mathrm{T}\boldsymbol{\Sigma}\boldsymbol{t}}\Re\left(\phi(\boldsymbol{t})\right)\right| > \epsilon\right) \le 2\exp\left(-\frac{n\epsilon^2}{2e^{2\boldsymbol{t}^\mathrm{T}\boldsymbol{\Sigma}\boldsymbol{t}}}\right),$$

$$\mathbb{P}\left(\left|\frac{1}{n}\sum_{j=1}^n e^{\boldsymbol{t}^\mathrm{T}\boldsymbol{\Sigma}\boldsymbol{t}}\sin\langle\boldsymbol{x}_j,\boldsymbol{t}\rangle - e^{\boldsymbol{t}^\mathrm{T}\boldsymbol{\Sigma}\boldsymbol{t}}\Im\left(\phi(\boldsymbol{t})\right)\right| > \epsilon\right) \le 2\exp\left(-\frac{n\epsilon^2}{2e^{2\boldsymbol{t}^\mathrm{T}\boldsymbol{\Sigma}\boldsymbol{t}}}\right).$$

Hence,

$$\mathbb{P}\left(\left|e^{\boldsymbol{t}^\mathrm{T}\boldsymbol{\Sigma}\boldsymbol{t}}\psi_n(\boldsymbol{t}) - \sum_{i=1}^K w_i\exp\left(\iota\langle\boldsymbol{\mu}_i,\boldsymbol{t}\rangle\right)\right| > \epsilon\right) = \mathbb{P}\left(\left|e^{\boldsymbol{t}^\mathrm{T}\boldsymbol{\Sigma}\boldsymbol{t}}[\psi_n(\boldsymbol{t}) - \phi(\boldsymbol{t})]\right| > \epsilon\right)$$

$$\le \mathbb{P}\left(\left|\Re\left(e^{\boldsymbol{t}^\mathrm{T}\boldsymbol{\Sigma}\boldsymbol{t}}[\psi_n(\boldsymbol{t}) - \phi(\boldsymbol{t})]\right)\right| > \frac{\epsilon}{\sqrt{2}}\right) + \mathbb{P}\left(\left|\Im\left(e^{\boldsymbol{t}^\mathrm{T}\boldsymbol{\Sigma}\boldsymbol{t}}[\psi_n(\boldsymbol{t}) - \phi(\boldsymbol{t})]\right)\right| > \frac{\epsilon}{\sqrt{2}}\right)$$

$$= \mathbb{P}\left(\left|\frac{1}{n}\sum_{j=1}^n e^{\boldsymbol{t}^\mathrm{T}\boldsymbol{\Sigma}\boldsymbol{t}}\cos\langle\boldsymbol{x}_j,\boldsymbol{t}\rangle - e^{\boldsymbol{t}^\mathrm{T}\boldsymbol{\Sigma}\boldsymbol{t}}\Re\left(\phi(\boldsymbol{t})\right)\right| > \frac{\epsilon}{\sqrt{2}}\right) + \mathbb{P}\left(\left|\frac{1}{n}\sum_{j=1}^n e^{\boldsymbol{t}^\mathrm{T}\boldsymbol{\Sigma}\boldsymbol{t}}\sin\langle\boldsymbol{x}_j,\boldsymbol{t}\rangle - e^{\boldsymbol{t}^\mathrm{T}\boldsymbol{\Sigma}\boldsymbol{t}}\Im\left(\phi(\boldsymbol{t})\right)\right| > \frac{\epsilon}{\sqrt{2}}\right)$$

$$\le 4\exp\left(-\frac{n\epsilon^2}{4e^{2\boldsymbol{t}^\mathrm{T}\boldsymbol{\Sigma}\boldsymbol{t}}}\right) \le 4\exp\left(-\frac{n\epsilon^2}{4e^{2\|\boldsymbol{t}\|_2^2\sigma_{\max}(\boldsymbol{\Sigma})}}\right) < \delta.$$

where we used $n > 4\log\left(\frac{4}{\delta}\right)\frac{e^{2\|\boldsymbol{t}\|_2^2\sigma_{\max}(\boldsymbol{\Sigma})}}{\epsilon^2}$ in the last inequality. $\square$

### E.1  PROOF OF THEOREM 2

*Proof.* Notice that

$$\left\|\mathrm{Proj}_{\boldsymbol{W}_k}\boldsymbol{X}\right\|_2^2 = \sum_{j=1}^n\left\|\mathrm{Proj}_{\boldsymbol{W}_k}\boldsymbol{x}_j\right\|_2^2 = \sum_{j=1}^n\sum_{l=1}^k|\langle\boldsymbol{x}_j,\boldsymbol{w}_l\rangle|^2.$$

Taking the expectation, we get

$$\mathbb{E}\left\|\mathrm{Proj}_{\boldsymbol{W}_k}\boldsymbol{X}\right\|_2^2 = \sum_{j=1}^n\sum_{l=1}^k\mathbb{E}|\langle\boldsymbol{x}_j,\boldsymbol{w}_l\rangle|^2$$

$$= \sum_{j=1}^n\sum_{l=1}^k\sum_{i=1}^K\mathbb{E}\left[|\langle\boldsymbol{x}_j,\boldsymbol{w}_l\rangle|^2|z_j = i\right]\mathbb{P}(z_j = i)$$

$$= \sum_{j=1}^n\sum_{l=1}^k\sum_{i=1}^K w_i\left(\sigma^2 + \mathbb{E}\left[\langle\boldsymbol{x}_j,\boldsymbol{w}_l\rangle|z_j = i\right]^2\right)$$

$$= \sum_{j=1}^n\sum_{l=1}^k\sum_{i=1}^K w_i\left(\sigma^2 + \langle\boldsymbol{\mu}_i,\boldsymbol{w}_l\rangle^2\right)$$

$$= \sum_{j=1}^n\left(k\sigma^2 + \sum_{i=1}^K w_i\sum_{l=1}^k\langle\boldsymbol{\mu}_i,\boldsymbol{w}_l\rangle^2\right)$$

$$= n\left(k\sigma^2 + \sum_{i=1}^K w_i\left\|\mathrm{Proj}_{\boldsymbol{W}_k}\boldsymbol{\mu}_i\right\|_2^2\right)$$

**Case 1:** $k = K$. We have

$$\mathbb{E}\left\|\text{Proj}_{\boldsymbol{W}_K}\boldsymbol{X}\right\|_2^2 \leq n\left(k\sigma^2 + \sum_{i=1}^K w_i\left\|\boldsymbol{\mu}_i\right\|_2^2\right) = \mathbb{E}\left\|\text{Proj}_{\boldsymbol{U}_K}\boldsymbol{X}\right\|_2^2,$$

where $\boldsymbol{U}_K = \text{span}\{\boldsymbol{\mu}_1, \cdots \boldsymbol{\mu}_K\}$.

**Case 2:** $k < K$. We show that the $k$-dimensional subspace $\boldsymbol{W}_k$ maximizing the $\mathbb{E}\left\|\text{Proj}_{\boldsymbol{W}_k}\boldsymbol{X}\right\|_2$ is the subspace of $\boldsymbol{U}_K$. Notice that

$$\sum_{i=1}^K w_i\left\|\text{Proj}_{\boldsymbol{W}_k}\boldsymbol{\mu}_i\right\|_2^2 = \sum_{i=1}^K\left\|\text{Proj}_{\boldsymbol{W}_k}\sqrt{w_i}\boldsymbol{\mu}_i\right\|_2^2$$

$$= \left\|\text{Proj}_{\boldsymbol{W}_k}\left[\sqrt{w_1}\boldsymbol{\mu}_1 \quad \cdots \quad \sqrt{w_K}\boldsymbol{\mu}_K\right]\right\|_2^2.$$

Therefore, the $k$-dimensional subspace maximizing the projection above is the subspace spanned by the first $k$ right eigenvectors of the SVD of $\left[\sqrt{w_1}\boldsymbol{\mu}_1 \quad \cdots \quad \sqrt{w_K}\boldsymbol{\mu}_K\right]$. This subspace $\boldsymbol{W}_k$ satisfies

$$\boldsymbol{W}_k \subseteq Im(\left[\sqrt{w_1}\boldsymbol{\mu}_1 \quad \cdots \quad \sqrt{w_K}\boldsymbol{\mu}_K\right]) = \boldsymbol{U}_K.$$

**Case 3:** $k > K$. We prove that the $k$-dimensional subspace $\boldsymbol{W}_k$ maximizing $\mathbb{E}\left\|\text{Proj}_{\boldsymbol{W}_k}\boldsymbol{X}\right\|_2$ must contain $\boldsymbol{U}_K$. Indeed, for any $\boldsymbol{W}_k$ such that $\boldsymbol{U}_K \subset \boldsymbol{W}_k$, we have

$$\mathbb{E}\left\|\text{Proj}_{\boldsymbol{W}_k}\boldsymbol{X}\right\|_2 = n\left(k\sigma^2 + \sum_{i=1}^K w_i\left\|\boldsymbol{\mu}_i\right\|_2^2\right).$$

$\square$

# F QUADRATIC PROGRAMMING OPTIMIZATION

In this section, we introduce the quadratic programming(QP) optimization applied in Algorithm 1 for recovering the component weights. The general formulation of the QP can be expressed as

$$\text{minimize } \frac{1}{2}\boldsymbol{x}^{\mathrm{T}}\boldsymbol{P}\boldsymbol{x} + \boldsymbol{q}^{\mathrm{T}}\boldsymbol{x} + r$$

$$\text{subject to } \boldsymbol{G}\boldsymbol{x} \preceq \boldsymbol{h}, \quad \boldsymbol{A}\boldsymbol{x} = \boldsymbol{b} \tag{34}$$

where $\boldsymbol{P} \in \mathbb{R}^{n \times n}, \boldsymbol{G} \in \mathbb{R}^{m \times n}, \boldsymbol{A} \in \mathbb{R}^{p \times n}$ and $\boldsymbol{P}$ is positive-definite. This optimization program can be viewed as minimizing a convex quadratic function over a polyhedron. For a more comprehensive introduction of the QP optimization, we refer to Boyd & Vandenberghe (2004).

## F.1 MIXING WEIGHTS ESTIMATION

After achieving the model order $\hat{K}$ and mean set $\{\hat{\boldsymbol{\mu}}_i : 1 \leq i \leq \hat{K}\}$, the corresponding weights are estimated by solving

$$\text{minimize } \left\|e^{-\boldsymbol{t}^{\mathrm{T}}\boldsymbol{\Sigma}\boldsymbol{t}}\sum_{i=1}^{\hat{K}} w_i e^{\iota\langle\hat{\boldsymbol{\mu}}_i,\boldsymbol{t}\rangle} - \psi_n(\boldsymbol{t})\right\|_2$$

$$\text{subject to } w_i \geq w, \quad \sum_{i=1}^{\hat{K}} w_i = 1 \tag{35}$$

The program (35) can be reformulated as a quadratic programming(QP) optimization and can be efficiently solved by well-established convex optimization toolboxes[2]. Next, we show how to fit the optimization problem (35) into the framework of (34). To simplify the notation, we replace $\hat{K}, \hat{\boldsymbol{\mu}}_i$'s with the unhatted ones. Notice that we can write

$$\left\|\sum_{i=1}^K w_i e^{\iota\langle\boldsymbol{\mu}_i,\boldsymbol{t}\rangle} - e^{\boldsymbol{t}^{\mathrm{T}}\boldsymbol{\Sigma}\boldsymbol{t}}\psi_n(\boldsymbol{t})\right\|_2^2 = \left\|\boldsymbol{A}\boldsymbol{\pi} - \boldsymbol{b}\right\|_2^2,$$

---

[2]In the numerical experiments, we use the python package cvxpy to implement the quadratic programming.

where $\boldsymbol{A} \in \mathbb{C}^{(N+1)^2 \times K}$ and $\boldsymbol{b} \in \mathbb{C}^{(N+1)^2}$ such that

$$A_{j,i} = e^{\iota \langle \boldsymbol{\mu}_i, \boldsymbol{t}_j \rangle}, \quad \boldsymbol{b}_j = e^{\boldsymbol{t}_j^{\mathrm{T}} \boldsymbol{\Sigma} \boldsymbol{t}_j} \psi_n(\boldsymbol{t}_j).$$

Here $\boldsymbol{t}_j$ is the $j$-th component of the vectorized sample points. Then the objective function can be further written as

$$\left\| \sum_{i=1}^{K} w_i e^{\iota \langle \boldsymbol{\mu}_i, \boldsymbol{t} \rangle} - e^{\boldsymbol{t}^{\mathrm{T}} \boldsymbol{\Sigma} \boldsymbol{t}} \psi_n(\boldsymbol{t}) \right\|_2^2 = \| \Re (\boldsymbol{A}\boldsymbol{\pi} - \boldsymbol{b}) \|_2^2 + \| \Im (\boldsymbol{A}\boldsymbol{\pi} - \boldsymbol{b}) \|_2^2$$

$$= \| \Re (\boldsymbol{A}) \boldsymbol{\pi} - \Re (\boldsymbol{b}) \|_2^2 + \| \Im (\boldsymbol{A}) \boldsymbol{\pi} - \Im (\boldsymbol{b}) \|_2^2$$

$$= (\Re (\boldsymbol{A}) \boldsymbol{\pi} - \Re (\boldsymbol{b}))^{\mathrm{T}} (\Re (\boldsymbol{A}) \boldsymbol{\pi} - \Re (\boldsymbol{b})) + (\Im (\boldsymbol{A}) \boldsymbol{\pi} - \Im (\boldsymbol{b}))^{\mathrm{T}} (\Im (\boldsymbol{A}) \boldsymbol{\pi} - \Im (\boldsymbol{b}))$$

$$= \boldsymbol{\pi}^{\mathrm{T}} [\Re (\boldsymbol{A})^{\mathrm{T}} \Re (\boldsymbol{A}) + \Im (\boldsymbol{A})^{\mathrm{T}} \Im (\boldsymbol{A})] \boldsymbol{\pi} - 2 [\Re (\boldsymbol{b})^{\mathrm{T}} \Re (\boldsymbol{A}) + \Im (\boldsymbol{b})^{\mathrm{T}} \Im (\boldsymbol{A})] \boldsymbol{\pi} + \Re (\boldsymbol{b})^{\mathrm{T}} \Re (\boldsymbol{b}) + \Im (\boldsymbol{b})^{\mathrm{T}} \Im (\boldsymbol{b}).$$

Therefore, we can fit (35) into the QP framework by setting

$$\boldsymbol{P} = \Re (\boldsymbol{A})^{\mathrm{T}} \Re (\boldsymbol{A}) + \Im (\boldsymbol{A})^{\mathrm{T}} \Im (\boldsymbol{A}), \quad \boldsymbol{q} = \Re (\boldsymbol{A})^{\mathrm{T}} \Re (\boldsymbol{b}) + \Im (\boldsymbol{A})^{\mathrm{T}} \Im (\boldsymbol{b}), \quad r = \frac{1}{2} \Re (\boldsymbol{b})^{\mathrm{T}} \Re (\boldsymbol{b}) + \frac{1}{2} \Im (\boldsymbol{b})^{\mathrm{T}} \Im (\boldsymbol{b})$$

in the objective function and setting

$$\boldsymbol{G} = -\boldsymbol{I}_K, \quad \boldsymbol{h} = -w \boldsymbol{1}_{K \times 1}, \quad \boldsymbol{A} = \boldsymbol{1}_{1 \times K}, \quad b = 1.$$

## G  THE EM ALGORITHM

In this section, we describe the EM algorithm used in the numerical experiments for comparison with our algorithms.

---

**Algorithm 5:** The EM algorithm (Fixed Covariance Matrix)

---

**input** : samples $\boldsymbol{x}_1, \cdots \boldsymbol{x}_n$, model order $k$, covariance matrix $\boldsymbol{\Sigma}$, initial guess $\hat{w}_i$'s, $\hat{\boldsymbol{\mu}}_i$'s

**1** *Expectation Step:* For $i = 1, \cdots, k$, compute

$$\gamma_i^j = \frac{w_i g(\boldsymbol{x}_i; \hat{\boldsymbol{\mu}}_i, \boldsymbol{\Sigma})}{\sum_{i=1}^{k} g(\boldsymbol{x}_i; \hat{\boldsymbol{\mu}}_i, \boldsymbol{\Sigma})}, \quad j = 1, \cdots, n.$$

**2** *Maximization Step:* Compute the weights and weighted means:

$$\hat{w}_i = \frac{1}{n} \sum_{j=1}^{n} \gamma_i^j, \quad \hat{\boldsymbol{\mu}}_i = \frac{\sum_{j=1}^{n} \gamma_i^j \boldsymbol{x}_j}{\sum_{j=1}^{n} \gamma_i^j}, \quad i = 1, \cdots, k.$$

**3** Iterate steps 1 and 2 until convergence.

---

In the numerical tests, we assume that the covariance matrix $\boldsymbol{\Sigma}$ is known as prior information. If the covariance matrix is unknown, it is updated in the maximization step by

$$\hat{\boldsymbol{\Sigma}} = \frac{\sum_{i=1}^{k} \sum_{j=1}^{n} \gamma_i^j (\boldsymbol{x}_j - \hat{\boldsymbol{\mu}}_i)(\boldsymbol{x}_j - \hat{\boldsymbol{\mu}}_i)^{\mathrm{T}}}{n}$$

for the unified covariance matrix case and

$$\hat{\boldsymbol{\Sigma}}_i = \frac{\sum_{j=1}^{n} \gamma_i^j (\boldsymbol{x}_j - \hat{\boldsymbol{\mu}}_i)(\boldsymbol{x}_j - \hat{\boldsymbol{\mu}}_i)^{\mathrm{T}}}{\sum_{j=1}^{n} \gamma_i^j}$$

for the general Gaussian mixture model.

