# OpenReview forum: "Learning High-dimensional Gaussian Mixture Models via a Fourier Approach"
_ICLR.cc/2025/Conference — ICLR 2025 Conference Withdrawn Submission_

### Official Review · Reviewer_nWTu · 2024-10-17

**Soundness:** 3
**Presentation:** 3
**Contribution:** 3
**Rating:** 6
**Confidence:** 3

**Summary:**

The authors study the problem of using iid
samples to estimate the D-dimensional Gaussian Mixture
Models (GMM)
parameters using an algorithm with linear complexity,
which challenges the current baselines (EM algorithm).

**Strengths:**

The paper studies theoretically the popular
problem of unsupervised learning of estimating the GMM
distribution of N input samples.
It states its goal clearly: it is about
the usage of Fourier operations, which lower the
complexity of the learning algorithm.
It uses concise language, it presents its model and
validates the complexity efficiency versus some variants
the EM algorithm.
It defers the detailed
discussion of MUSIC for the appendix.
The above points make it well written
and clearly presented.

From a first study, the model seems mathematically
sound and the suggested Fourier operations
form a learning algorithm which seems to be novel.

The assumptions of the model are restrictive: the
authors acknowledge that in Lines 530 - 538.

**Weaknesses:**

The paper chooses to analyze the complexity of the
model and avoids allocating a discussion
in practical applications of GMM estimation. That
would make it stronger as a machine learning study.

**Questions:**

In the 'Primary Area' section, the authors
chose to classify their paper in
'semi-supervised, and supervised representation learning'. For the N iid input samples,
is there any vector of known classes (i.e. ground truth)
used in their estimation algorithm?

Can the authors provide some applications
of their algorithm in these two areas?

---

### Official Review · Reviewer_VYt1 · 2024-10-30

**Soundness:** 2
**Presentation:** 2
**Contribution:** 2
**Rating:** 3
**Confidence:** 3

**Summary:**

The paper provides a lower bound on sample complexity for learning mixture of Gaussians.

It proposes a PCA based approach to solve learning mixture of Gaussians, which is comparable to EM.

**Strengths:**

The paper provides sample complexity lower bounds.

**Weaknesses:**

I couldn't find theoretical guarantees for the proposed algorithm.

**Questions:**

Do we know if the proposed algorithm can achieve the lower bound?

---

### Official Review · Reviewer_FB2n · 2024-11-02

**Soundness:** 2
**Presentation:** 1
**Contribution:** 4
**Rating:** 5
**Confidence:** 4

**Summary:**

The paper considers the problem of estimating the means and the number of components of the GMM from samples. It proposes a new approach based on estimating the characteristic function (based on Fourier transform) and characterize the number of samples and time required by the algorithm to estimate the means and the order of the GMM.

**Strengths:**

The paper proposes a new algorithm for GMMs and characterizes its complexity. The results are novel and would be of interest to researchers in the field.

**Weaknesses:**

My biggest concern about the paper is that the current writing assumes the reader is familiar with many technical terms which are not commonly used in machine learning or TCS conferences, for example terms such as 'spectral estimation algorithm', 'stable recovery', 'cutoff frequency', which are stated and used without being formally defined. While researchers in statistics may be familiar with these terms, the paper is difficult to follow for ML theory researchers. The paper also has some typos or mistakes (see the questions section), which makes it hard to follow. Here are few suggestions which might improve the flow (in no specific order):

Some notations are used before they are defined, e.g.,
- Line 136: D is not defined at this point.
- Line 143: w_{min} and \Delta are not defined yet.
Line 142, 144: what is "stable recovery" and "Stable estimation"? Does it mean with high probability or something more subtle? It might be good to formally define it.
Line 233, 249: what is L?
Line 250: what is \tilde{K}?
Line 377: what is M?

Some parts can use a reference
- Line 192: why is the variance of asymptotic normality ( 1- |\phi(t)|^2)? Please add a citation.
- Line 246: why does Nyquist-Shannon sampling theorem imply that to estimate the mean, one needs h < \pi/R?

Some terms may not be easily understood by ICLR researchers
- Line 194: why is multiplying by e^{t^T \Sigma t} considered as modulating? While this is not critical to the story, there are many terms used like this, which may not be familiar with readers.

I believe due to space constraints, the MUSIC algorithm is provided in the Appendix, while this is fine, it might be good to have some explanation of the algorithm in the main paper.

Line 236: Algorithm 1: what is the difference between w and \pi? Is there a reason to use \hat{w}_i to denote an estimate of \pi_i?

Line 285: what is \epsilon(t)? in Equation 13? Since this is a fixed quantity, by display equation in line 197, isn't y(t) = modulated characteristic function when epsilon(t) = 0? Also what does the assumption ||epsilon(t) ||_\infty \leq sigma mean?

Remarks in line 320 and 324: The abstract currently suggests the provided bounds are the min-max rates for order estimation. However, after reading these two remarks, I realized that these are rates for a specific algorithm.  I believe this confusion is not intentional, but would be good to clarify.

**Questions:**

- Please see the weakness question above for questions regarding definitions and nomenclature?

- Line 210: I believe the last term should have \sigma_max(\Sigma), rather than \sigma_min(\Sigma). Note that t^{T} \Sigma T > ||t|^2 \sigma_{\min}(\Sigma) and hence 1/(t^{T} \Sigma T) < 1 /(||t|^2 \sigma_{\min}(\Sigma)) and hence -1/(t^{T} \Sigma T) > -1 /(||t|^2 \sigma_{\min}(\Sigma)). This changes the result in line 214, so the dependence on n also changes to sigma_max. Would it change any other results in the paper? For example, would it change Equation 15?

Line 318: In theorem 1, it is stated that \Delta >= R_{D, K} holds with probability at least 1 - delta. From Definition 2, \Delta seems to be a property of the underlying distribution and does not have anything to do with samples. So I am not sure what this theorem means.

---

### Official Review · Reviewer_QUAn · 2024-11-02

**Soundness:** 2
**Presentation:** 2
**Contribution:** 2
**Rating:** 3
**Confidence:** 4

**Summary:**

This paper proposes two algorithms for estimating parameters in Multivariate Gaussian Mixture Models with a known common covariance matrix. The first algorithm leverages the MUSIC (Multiple Signal Classification) algorithm, along with a quadratic program, to simultaneously determine the model order $K$, estimate mean parameters, and estimate the mixing weights $w_i$. For small problem dimensions $d$, the authors propose gridding a bounded region in $R^d$ using $N^d$ data points to approximate an appropriate Fourier function, which is then used to determine the model order and parameters. For high-dimensional problems, the authors employ a Principal Component Analysis-based approach to first perform dimensionality reduction, and then apply the gridding technique (the first algorithm) to estimate the parameters.

**Strengths:**

The paper addresses the important problem of Gaussian mixture model estimation, with particular emphasis on determining the unknown number of mixture components. The authors propose a Fourier series-based method designed to simultaneously extract both the number of components $K$ and the model parameters $\mu_i's$ and $w_i's$. Through extensive simulations, the authors demonstrate that their proposed algorithm performs comparably to or better than the traditional EM algorithm.

**Weaknesses:**

The paper, while well motivated, lacks concrete theoretical guarantees. For instance:

1. It is not clear how accurately the number of components $K$ of the model can be estimated by the MUSIC algorithm (even in small dimensions in presence of noise). More concretely, is the estimate (say K-hat) for the number of components a consistent estimator for the  true number of components $K$?  What happens to this estimate when the dimension of the problem is large.



2. Are the estimates of weights and mean vectors are consistent? The authors suggest to run  their algorithm with a larger number of components when the true number of components are unknown (page 14 lines 752-755). It is not clear what happens in that scenario. To best of my under standing, one selling point of the paper (compared to K-means or the EM algorithm) was that it detects the number of components accurately. It is not clear from the presentation of the paper, how justified is this claim.



3. Finally, the paper recommends to grid a bounded region in the space $R^d$ using $N^d$ data points, and hence it is not efficient even when $d$ is small. On the contrary, the competing algorithms like K-means and EM-algorithm do not face such issues.



4. The paper seems to be an extension of the MUSIC algorithm for one dimension [1]. It would be nice to know what are some technical challenges that are novel in this setting.

[1]. Xinyu Liu and Hai Zhang. A fourier approach to the parameter estimation problem for one-
dimensional gaussian mixture models. arXiv preprint arXiv:2404.12613, 2024.

**Questions:**

Please look at the weakness section.

---

### Official Review · Reviewer_ACfx · 2024-11-06

**Soundness:** 4
**Presentation:** 4
**Contribution:** 4
**Rating:** 6
**Confidence:** 4

**Summary:**

In this paper, the authors tackle the problem of parameter estimation from a GMM with iid samples when even the number of components is unknown. To do so, the authors exploit Fourier measurements of the samples - a novel technique in itself and proposes an algorithm with linear time complexity. The authors further show the use of PCA in dimension reduction for improved computational guarantees.

**Strengths:**

The paper is well written. Both the problem and the techniques used in the paper are novel and relevant.

**Weaknesses:**

I have some questions regarding this work

1. I am worried that instead of the number of components being a hyperparameter, we now have the cutoff frequencies as a hyperparameter to the algorithm. Why is the latter better than the former? What is L in Algorithm 1?

2. It is critical that the MUSIC algorithm is inserted into the main text. Otherwise the reader who is not acquainted with it is not able to understand it at all.

3. The algorithm is very poorly written. The function \Psi_n(t) is not referred to appropriately. In Step 2, what is \mu? What is the quadratic program written in Step 4? The entire algorithm is a black box

4. There is no intuition provided for the MUSIC algorithm.

5. The lower bound in Remark 324-330 talks about the sample complexity when components is known - is that a contribution of this work? In any case, for K=5 and Delta=1/2, the dependence in lower bound is 2^{18}. On the other hand, the upper bound in Theorem 1 has dependence of 2^{16}. Why is the upper bound smaller than the lower bound? What am I missing?

5) The fact that PCA reduced "time" complexity needs to highlighted appropriately - confusion with sample complexity in several places

**Questions:**

Please see above

---

### Official Review · Reviewer_sLdb · 2024-11-09

**Soundness:** 2
**Presentation:** 2
**Contribution:** 2
**Rating:** 3
**Confidence:** 3

**Summary:**

The paper proposes an algorithm for estimating the means, the order (number of components), and the mixing distribution (the distribution over components) of a Gaussian mixture model through the measurements in the fourier space of the empirical measure of independent samples. Assuming the knowledge of the covariance, the work provides lower-bounds on the sample-complexity for estimating the order and mixing distribution. The paper further suggests applying PCA to improve the algorithm's computational complexity by estimating the low-dimensional subspace spanned by the means. Lastly, the paper numerically evaluates the proposed approach against EM (Expectation Maximization) baselines and criteria for estimation of the model order.

**Strengths:**

- The paper does an adequate job at describing the problem setup and the proposed algorithm.
- The proposed approach could have utilities in applications with known low-frequency bias in the fourier domain.
- Numerical results suggest improved computational efficiency over the EM baseline with similar performance.

**Weaknesses:**

- The theoretical analysis in the work is limited. The paper only provides lower bounds on the sample-complexity for estimating the model order and mixing distribution without a discussion of the sample-complexity for recovering the means themselves. The paper doesn't describe the conditions under which the mixture means can be estimated through the local minima of the MUSIC imaging function apart from lower-bounds on the necessary discretization. The PCA-based result is also directly adapted from Vempala & Wang 2002 and is not combined with the analysis for the main algorithm. Ideally, the sample-complexity guarantees should be compared with information-theoretic/minimax-optimal lower-bounds at a given scaling of separation distance and the covariance.
- The proposed algorithm is an extension of Liu and Hai Zhang 2024 to multi-dimensional random variables and therefore possesses limited novelty.
- It's unclear what assumptions on the fourier decomposition of the covariance are assumed for the proposed algorithm to be sample-complexity efficient. Even for the sample-complexity lower bounds for estimating the order, the dependence on the fourier decomposition of the covariance is hidden in the quantity $f$. It should be clarified how $f$ scales under a scaling of $\Delta$ and $\Sigma$ which allows recovery.

**Questions:**

- What are the theoretical guarantees for recovering the means through the ``largest k maxima of $\mathcal{J}$" in the MUSIC algorithm?
- What is L in algorithm 1 and section 2.2?
- From eq. 17, each $f_d$ can in general be $\Theta(1)$. As per the definition of f in eq. 13, doesn't this imply an exponential dependence on the dimension in the sample complexity given by eq. 15?
- In Theorem 2, what is the requirement on n?
- Why can't the algorithm in Vempala & Wang (2002) be further used to directly estimate the means of the mixtures? How does this compare to the MUSIC algorithm?

-- Typos:
- Line 388: expectated -> expected

---

### Note · Authors · 2024-12-03

I have read and agree with the venue's withdrawal policy on behalf of myself and my co-authors.